# ViLMA: A Zero-Shot Benchmark for Linguistic and Temporal Grounding in Video-Language Models

**Ilker Kesen**[1,2][*]    **Andrea Pedrotti**[3,4]    **Mustafa Dogan**[5,6]    **Michele Cafagna**[7]
**Emre Can Acikgoz**[1,2]    **Letitia Parcalabescu**[8]    **Iacer Calixto**[9,10]    **Anette Frank**[8]
**Albert Gatt**[7,11]    **Aykut Erdem**[1,2]    **Erkut Erdem**[1,5]

[1] Koç University, KUIS AI Center  [2] Koç University, Department of Computer Engineering
[3] University of Pisa, Department of Computer Science
[4] Institute of Information Science and Technologies, Italian National Council of Research
[5] Hacettepe University, Department of Computer Engineering  [6] Aselsan Research
[7] University of Malta, Institute of Linguistics and Language Technology
[8] Heidelberg University, Department of Computational Linguistics
[9] Amsterdam UMC, University of Amsterdam, Department of Medical Informatics
[10] Amsterdam Public Health, Methodology & Mental Health, Amsterdam, The Netherlands
[11] Utrecht University, Department of Information and Computing Sciences

## Abstract

With the ever-increasing popularity of pretrained Video-Language Models (VidLMs), there is a pressing need to develop robust evaluation methodologies that delve deeper into their visio-linguistic capabilities. To address this challenge, we present ViLMA[1]), a task-agnostic benchmark that places the assessment of fine-grained capabilities of these models on a firm footing. Task-based evaluations, while valuable, fail to capture the complexities and specific temporal aspects of moving images that VidLMs need to process. Through carefully curated counterfactuals, ViLMA offers a controlled evaluation suite that sheds light on the true potential of these models, as well as their performance gaps compared to human-level understanding. ViLMA also includes proficiency tests, which assess basic capabilities deemed essential to solving the main counterfactual tests. We show that current VidLMs' grounding abilities are no better than those of vision-language models which use static images. This is especially striking once the performance on proficiency tests is factored in. Our benchmark serves as a catalyst for future research on VidLMs, helping to highlight areas that still need to be explored.

## 1 Introduction

Video-language models (VidLMs) have received increasing attention from the research community (Lei et al., 2021; Luo et al., 2022; Xu et al., 2021; Zellers et al., 2021; Luo et al., 2020; Fu et al., 2021; Ma et al., 2022; Bain et al., 2021; Ge et al., 2022; Lei et al., 2022; Zhu et al., 2022; Cheng et al., 2023). In principle, VidLMs can visually ground linguistic phenomena which are beyond the reach of image-language models (ILMs),[2] since videos include *dynamically evolving phenomena* (e.g., events, actions, physical processes). Nonetheless, this *temporal dimension* makes learning more complex. Most efforts to gauge what VidLMs can do rely on *tasks* such as video captioning (Yu et al., 2016), text-to-video retrieval (Wang et al., 2021), and video question answering (Yu et al., 2019). While such evaluations shed light on task performance and support comparative analysis, they are limited in their ability to reveal the specific visuo-linguistic capabilities that models exhibit *across tasks*.

---

[*]Corresponding author. Email: ikesen16@ku.edu.tr
[1]Project page: `https://cyberiada.github.io/ViLMA`
[2]Image-language models are trained on images and text, and have shown strong performance on many tasks (Mogadala et al., 2021; Du et al., 2022; Agrawal et al., 2022; Chen et al., 2023).

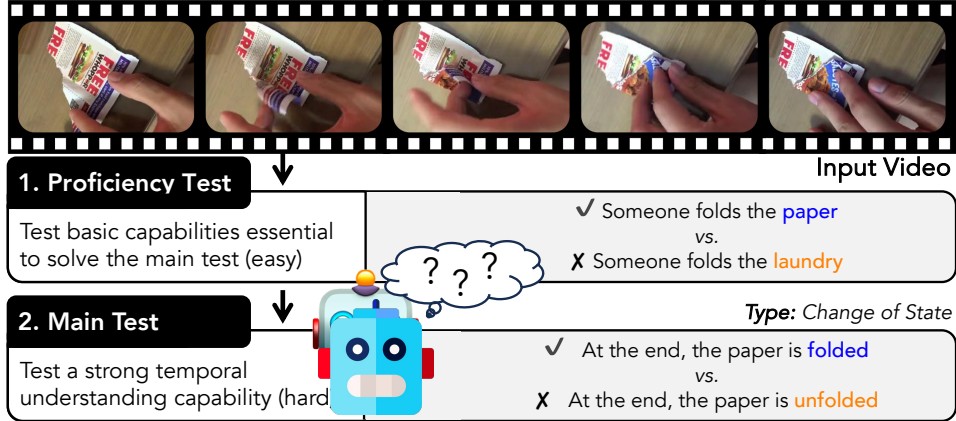

Figure 1: An overview of VILMA. A *proficiency test* first evaluates basic understanding skills of a model, followed by a more complex *main test* for a specific temporal reasoning capability.

In this study, we present VILMA (Video Language Model Assessment), a task-agnostic benchmark that proposes a behavioural evaluation for VidLMs focusing on fine-grained phenomena. We draw inspiration from related benchmarks for ILMs (e.g. Parcalabescu et al., 2022; Hendricks & Nematzadeh, 2021; Thrush et al., 2022). However, VILMA focuses on tests that require *strong temporal understanding and reasoning*, as time is a unique aspect present in VidLMs but not in ILMs. We adopt a common structure for each *test*: (i) We harvest high-quality examples from existing video-language datasets; (ii) we create counterfactual examples or 'foils' (Shekhar et al., 2017b), so that a *test* requires distinguishing correct from counterfactual video+text pairs; (iii) we create a *proficiency test* to gauge if a model learns the capabilities we deem necessary to solve the main test; (iv) we apply automatic and manual validation of the examples and their counterfactuals to control for biases and to ensure a high-quality evaluation benchmark; (v) finally, we test whether existing VidLMs can solve the proficiency tests and distinguish correct from counterfactual examples in the main tests (see Figure 1). Our main contributions can be listed as follows:

- We propose VILMA, a zero-shot benchmark for evaluating VidLMs, designed to require strong *temporal understanding*. To the best of our knowledge, this is the first behavioural benchmark to test VidLMs for temporal visuo-linguistic capabilities.

- We devise a *proficiency test* for each *main test* in our benchmark, to probe for basic capabilities we deem essential for solving the task correctly.

- We report experiments that demonstrate the usefulness of VILMA to evaluate VidLMs on different criteria. In particular, our results also show that current VidLMs are not significantly better at temporal reasoning than ILMs.

- We show that accounting for proficiency tests leads to a significant decrease in performance, suggesting that many apparently correct predictions by VidLMs could be accidental or spurious.

The rest of this paper is structured as follows: In §2, we briefly review the relevant literature. In §3, we describe our data generation methodology in detail. In §4, we report our experimental setup and results. Finally, in §5, we summarise our conclusions.

## 2 RELATED WORK

In this section, we categorise pretrained video-language models (VidLMs) (§2.1), review recent efforts that investigate the capabilities of pretrained image-language models (ILMs) (§2.2), and position our work in relation to existing video-language benchmarks (§2.3).

## 2.1 PRETRAINED VIDLMS

We categorise VidLMs along five distinct dimensions: modality considered for pretraining, pretraining datasets, pretraining objectives, strategies for temporal modelling and multi-modal fusion schemes. See §4 for detailed descriptions of models used in our experiments.

**Modalities.** Pretraining of VidLMs can be performed on images (Lei et al., 2021), videos (Li et al., 2020; Zhu & Yang, 2020; Xu et al., 2021; Zellers et al., 2021; Seo et al., 2022; Wang et al., 2022a; Li et al., 2022a; Luo et al., 2022) or both (Bain et al., 2021; Fu et al., 2021; Wang et al., 2022b; Li et al., 2022c; Lei et al., 2022; Li et al., 2023b). A handful of models (Akbari et al., 2021; Lin et al., 2022; Zellers et al., 2022) also incorporate speech and audio.

**Datasets.** Training data is often chosen in view of the type of pretraining used for the visual modality. Early VidLMs (Zhu & Yang 2020; Li et al. 2020; Xu et al. 2021) use HowTo100M (Miech et al., 2019), which offers the linguistic modality in form of automatic speech recognition (ASR) output or manually written subtitles. Recent models are pretrained on the WebVid-2M dataset (Bain et al., 2021), which follows a similar approach to Conceptual Captions (CC3M; Sharma et al., 2018) in filtering items based on the quality of the textual modality. Next to video-text data, recent VidLMs also leverage large image-text datasets, e.g. SBU captions (Ordonez et al., 2011), CC3M or CC12M (Changpinyo et al., 2021).

**Objectives.** Some pretraining objectives for VidLMs have been derived from the pretraining objectives employed by ILMs. The most prominent among these are video-text contrastive loss (VTC), video-text matching (VTM), masked language modelling (MLM) and masked frame modelling (MFM). A few models employ natural language generation (NLG) (Seo et al., 2022; Wang et al., 2022b), masked visual-token modelling (MVM) (Li et al., 2022c), or temporal reordering (Zellers et al., 2021).

**Temporal Modelling.** Only a few methods use joint space-time attention (Bertasius et al., 2021; Bain et al., 2021; Wang et al., 2022b) to process video. Some approaches (Zellers et al., 2021; Luo et al., 2022; Yang et al., 2022) rely on language at this stage, and implement a multi-modal attention mechanism between patches and word embeddings. Fu et al. (2021); Li et al. (2022c) extract spatio-temporal features using the Video Swin Transformer (Liu et al., 2022) with shifted window attention (Liu et al., 2021).

**Multi-modal Fusion.** Models relying exclusively on the VTC objective do not perform multi-modal fusion (Xu et al., 2021; Bain et al., 2021; Luo et al., 2022; Lin et al., 2022). Others either include an additional multi-modal transformer (Luo et al., 2020; Lei et al., 2022; Seo et al., 2022) or fuse a visual prefix into text-only LMs (Zellers et al., 2021; Fu et al., 2021).

## 2.2 BENCHMARKS FOR PRETRAINED IMAGE-LANGUAGE MODELS (ILMS)

ILMs are usually tested on downstream *tasks* such as image question answering (Goyal et al., 2017b), visual reasoning (Suhr et al., 2019) or image retrieval (Lin et al., 2014; Plummer et al., 2015). Some benchmarks measure *task-overarching capabilities* of ILMs (e.g., their understanding of verbs; Hendricks & Nematzadeh, 2021), or compositionality (Thrush et al., 2022). A specific way of testing ILMs is *foiling* (Shekhar et al., 2017b; Gokhale et al., 2020; Bitton et al., 2021; Parcalabescu et al., 2021; Rosenberg et al., 2021), where a caption is turned into a counterfactual (i.e., *foil*) by minimal edits, such that it does not correctly describe the image anymore (Shekhar et al., 2017b;a). Alternatively, the image can be exchanged such that it does not match the caption anymore (Rosenberg et al., 2021; Wang et al., 2023). A key consideration in creating counterfactuals is to target specific linguistic elements, which are assumed to reflect specific model capabilities (e.g. by altering a preposition, a model's ability to distinguish caption from foil should reflect its understanding of spatial relations). For example, the VALSE benchmark (Parcalabescu et al., 2022) tests the linguistic grounding capabilities of ILMs targeting six linguistic phenomena: existence, plurality, counting, spatial relations, actions, and entity coreference. ILMs are tested zero-shot on image-text alignment, one of the ILM's pretraining objectives.

An alternative strategy is to test pretrained models on multiple choice questions designed to probe specific capabilities (cf. the recent SEED-Bench Li et al., 2023c).

Bugliarello et al. (2023) tested recent encoder-only ILMs on several benchmarks mentioned above: SVO probes (Hendricks & Nematzadeh, 2021), VALSE, and Winoground (Thrush et al., 2022).

## 2.3 BENCHMARKS FOR PRETRAINED VIDLMS

Like ILMs, VidLMs are evaluated on numerous downstream tasks, primarily action recognition (Kuehne et al., 2011; Soomro et al., 2012), video-text retrieval (Xu et al., 2016; Hendricks et al., 2017), and video question answering (VidQA) (Xu et al., 2017; Lei et al., 2018). Lei et al. (2022) show that a non-temporal model can perform better than temporal models in these benchmarks. Newer VidQA benchmarks (Lei et al., 2020; Xiao et al., 2021) offer stronger tests for VidLMs to probe their temporal and commonsense reasoning capabilities. In our benchmark, we also prioritise these aspects. However, we cast the tasks in a zero-shot setting using a counterfactual setup, to probe the pretrained models' inherent capabilities.

Foiling benchmarks have also been proposed to evaluate VidLMs. Park et al. (2022) devise two tests. In the first one, foils are created by swapping the character entities in the caption. In the second, an LM replaces the verb phrase of the caption. On the other hand, Bagad et al. (2023) create a benchmark consisting of synthetic video-caption-foil triplets (e.g. a red circle appears after/before a yellow circle) to test how well VidLMs localise the events happening in the video. Bagad et al. (2023) also propose a *consistency* test to probe whether the models localise the events correctly or just predict the correct answers. One of the tasks in VILMA is similar to Park et al. (2022), but we build it upon the Situation awareness task, which tests for models' ability to reason about actors, actions, and their relationships (see § 3.3). Similar to the consistency task of Bagad et al. (2023), we propose a *proficiency test* for each of our main tests. In contrast to earlier foiling benchmarks, VILMA is also more comprehensive as it is designed to examine the models' grounding capabilities for different linguistic phenomena.

Another notable benchmark is VALUE (Li et al., 2021). VALUE follows the design of the (Super) GLUE evaluation suites (Wang et al., 2019a;b) for NLU, offering 11 datasets covering 3 different downstream tasks. Unlike VALUE, VILMA is a zero-shot *foiling* benchmark with particular focus on linguistic phenomena that emphasise temporal reasoning.

## 3 CONSTRUCTING VILMA

VILMA is designed as a *probing benchmark* divided into five main tests, summarised in Table 1 and described in detail below. It is intended as a zero-shot evaluation benchmark. For each test, we define *specific foiling functions* that target central characteristics of VidLMs, focusing on their *temporal understanding capabilities*.

First, we introduce *proficiency tests* (§3.1). They test criteria that can be considered as *prerequisites for solving the main tests*, by assessing the VidLMs' capability to successfully navigate and solve simpler objectives before attempting the more demanding main tests. We then introduce our main tests, which focus on: accurately recognising events that display *temporal regularity/periodicity and recurrence*, i.e., action counting (§3.2); the recognition of specific *actions or action participants* (§3.3); the recognition of *action or event subphases*, especially when they induce a change of state (§3.4); the influence of model biases and frequency effects in VidLM's understanding of *rare actions* (§3.5); and distinguishing *spatial relations* (§3.6), since these often exhibit temporal evolution (e.g. in the case of an object moving *towards* another) and thus alter in their visual appearance over time. Finally, in §3.7 we discuss how we use human validation to guarantee VILMA's quality.

### 3.1 PROFICIENCY TESTS

Proficiency tests can be considered a preliminary criterion for each of the five main tests below. These tests assess a VidLM's ability to solve simpler visuo-linguistic tasks that do not require strong temporal modelling, as the main tests do. In contrast, VidLMs are expected to address the primary tests by effectively modelling temporal dynamics. Consequently, foils in the proficiency test are less challenging compared to the main tests, and serve as an additional evaluation criterion.

The rationale behind conducting proficiency tests is as follows: When a model can effectively tackle the main test but falls short of passing its corresponding proficiency test, it raises a crucial point of

Table 1: Overview of data and foiling methods used in each test in VɪLMA.

| Test (#exs.) | Video Caption (blue) / Foil (orange) | Foil Generation | Sample Frames |
|---|---|---|---|
| **Action Counting** (1432) | Someone lifts weights exactly two / five times. | Number replacement |  |
| **Situation Awareness** (911) | A policeman / blond man holds a blond man / policeman against a wall. | Actor swapping |  |
| | A man in blue holds / chops up a man in green. | Action replacement | |
| **Change of State** (998) | Someone folds / unfolds the paper. | Action replacement |  |
| | Initially, the paper is unfolded / folded. | Pre-state replacement | |
| | At the end, the paper is folded / unfolded. | Post-state replacement | |
| | Initially, the paper is unfolded / folded. Then, someone folds / unfolds the paper. At the end, the paper is folded / unfolded. | Swap-and-replacement | |
| **Rare Actions** (1443) | Drilling into / Calling on a phone. | Action replacement |  |
| | Drilling into a phone / wall. | Object replacement | |
| **Spatial Relations** (393) | Moving steel glass towards / from the camera. | Relation replacement |  |

concern. This discrepancy hints that the VidLM may potentially be relying on heuristics that exploit biases inherent within the modalities. These biases, in turn, should presumably be traced back to the early pretraining phase of the models.

Given the individual characteristics of the tests, the proficiency test focuses on specific objectives in each case: For the Spatial Relations (§3.2), Change of State (§3.4), and Situation Awareness (§3.3) tests, the aim of the proficiency test is to **identify objects** mentioned in the captions. On the other hand, in the Action Counting (§3.2) and Rare Actions (§3.5) tests, we shift our attention to **action recognition** and **object existence**, respectively.

We use SpaCy's[3] dependency parser to localise and mask the target words. These words are then replaced with foil words generated via Masked Language Modelling (MLM)[4]. To ensure the validity of our proficiency tests we rely on manual evaluation as well as further constraints in the creation process. For the details we refer readers to Appendix C.1.

## 3.2 ACTION COUNTING

The **Action Counting** test probes the ability of models to accurately count the occurrences of actions within a given video input stream. This test requires *spatio-temporal reasoning*, presenting a novel and interesting challenge. To this end, we use the QUVA dataset (Runia et al., 2018), which comprises 100 videos. Within each video, every occurrence of the target action is annotated with a corresponding frame number that specifies the end of each action.

The dataset lacks any textual annotations. Consequently, we curate multiple textual templates per video, incorporating a placeholder for the numerical value (<number>). Our templates incorporate

---

[3] https://github.com/explosion/spaCy
[4] We use RoBERTa-large to fill the mask token in the modified captions with the most contextually appropriate token.

the term *exactly* to indicate precise counting (e.g., someone performs exactly `<number>` push-ups); cf. Parcalabescu et al. (2022) for a similar strategy. We avoid overly specific terms, opting for more general descriptors (e.g., *lifting weights* instead of *skull-crushers arm exercise*). A native English speaker checked the manually curated templates and fixed potential syntax errors in them.

We replace the number placeholder with the correct numerical value to create captions, and with an incorrect one to create foils. We discard all instances with counts exceeding a predetermined threshold $T_c$, set at 10. For the counting test, we created the following two subtests: In the **Easy** subtest, we deliberately opt for small numbers $C \in \{1, 2, 3\}$ in the captions. The choice of these small numbers is motivated by the observation that models frequently encounter such quantities during pretraining, making them more likely (and possibly more easily recognisable). In the **Difficult** subtest, by contrast, we favour these same small numbers in the foils. This presents a challenge for VidLMs as it tests the models' ability to overcome biases towards numbers frequently present in pretraining. In this way, we aim to assess the models' true abilities to handle counting tasks in diverse contexts. We describe our data collection process in detail in Appendix C.2.

## 3.3 SITUATION AWARENESS

The **Situation Awareness** test shows how effectively VidLMs grasp the interaction between visual clues and verbal context by testing whether they recognise actors, actions, and their relationships. To this end, we use the VidSitu (Sadhu et al., 2021) dataset consisting of 10-second video sequences annotated with information regarding verbs, semantic roles, entity co-references, and event relations. To add captions to this dataset, we use ChatGPT to refine and enhance the template-based sentences generated from the existing annotations.

Unlike tests which target verb-argument structure in ILMs, such as SVO-Probes Hendricks & Nematzadeh (2021) and the verb replacement and actant swap tests in VALSE (Parcalabescu et al., 2022), this video-language task adds a temporal dimension, encapsulating dynamic actions. Unlike static images, videos illustrate unfolding events and track their temporal dynamics via sequences of frames. VidLMs must grasp frame coherence, temporal context, and story structure, assessing the order of occurrences. In contrast, ILMs focus on static imagery with less temporal emphasis. Furthermore, videos introduce audio and motion, which gives the current task broader scope and presents novel challenges for contextual integration.

Our Situation Awareness test consists of the **Action Replacement** and **Actor Swapping** subtests. **Action Replacement** tests whether VidLMs can distinguish various activities, by contrasting phrases that differ only in action verbs. To that end, we mask the verb in a caption with a `<MASK>` token and generate foils via masked language modelling. Subsequently, we employ natural language inference (NLI) filtering to validate the foils, using an ALBERT model (Lan et al., 2020). We only consider foils that are predicted as 'contradiction' or 'neutral' with respect to the original caption by the NLI model. Finally, we compute a grammaticality score for all foils using GRUEN (Zhu & Bhat, 2020) and only retain as valid cases where the GRUEN grammatically exceeds 80%.

**Actor Swapping** tests the VidLMs' ability to recognise the role played by (human) actors in diverse actions, thereby probing the ability to discern the semantic roles of arguments in complex relations. To generate foils for the **Actor Swapping** subtest, we interchange the action participants in a caption. We do not apply NLI or GRUEN grammatically filters. Please refer to Appendix C.3 for further details on the construction of this test.

## 3.4 CHANGE OF STATE

The **Change of State** test examines the ability of VidLMs (i) to recognise and distinguish different sub-phases of actions, especially those that induce a *change of state (CoS)* of objects or entities involved in it; and (ii) to align the beginning and ending phases of these actions across modalities. Cross-modal alignment of the begin- and end-states of CoS actions is challenging, as they are typically *textually implicit* while being *visually explicit*.

We define as *CoS verbs* those verbs that refer to actions that include (or textually imply) an initial situation (or state) that is modified to an outcome situation (or state) (e.g., *"to open (a bottle)"* implies that an initial state of *"(the bottle) being closed"* changes to an outcome state of *"(the bottle) being*

*open"* as a result of an opening action). We further assume that the outcome must differ from the initial state.

We collect our target *CoS verbs* starting from a codebase by Warstadt et al. (2019). While the authors only provide the initial-state for each verb, we expand the list by identifying appropriate outcomes for all actions. Leveraging the list of *CoS verbs* as targets, we collect candidate sentence-video pairs by parsing various multimodal datasets: Something-Something V2 (Goyal et al., 2017a), YouCook2 (Zhou et al., 2018), COIN (Tang et al., 2019), RareAct (Miech et al., 2020), and STAR (Wu et al., 2021). We extract the subject and object from the collected sentences, and generate a caption according to a pre-defined template. We generate foils by replacing one or more sub-phases (action, pre-state or post-state) with their respective opposite expressions.

We design four different subtests, in each of which we foil an expression describing a specific element: **Action** subtest, **Pre-state** subtest, **Post-state** subtest, and **Reverse** subtest, where we swap pre-state and post-state and replace the action with its antonym. This reverses the original linguistic sequence, e.g. turning 'closed–*open*-open' to 'open-*close*-closed', which serves as a linguistically coherent foil for the original action in the video. For more details, please see Appendix C.4.

### 3.5 RARE ACTIONS

The **Rare Actions** test probes how well VidLMs identify novel compositions and recognise unusual interactions between human beings and objects. We leverage the RareAct dataset (Miech et al., 2020) consisting of videos accompanied by action-object pairs describing events within the videos. These action-object pairs are extracted by analysing co-occurrence statistics from the widely used HowTo100M (Miech et al., 2019) dataset.

To enrich this dataset, we generate simple captions based on the action-object pairs. For instance, given the action-object pair *cut-keyboard*, we create the descriptive caption *cutting a keyboard*. This test offers two subtests: In **Action Replacement**, we substitute the original action with a more plausible alternative that can be applied to the given object, e.g. *type on* for the previous *keyboard* example. To generate foils in this subtest, we employ T5 (Raffel et al., 2020), as it enables us to produce foils with *phrasal verbs*, e.g., *talk about*, *place at*, etc. As for **Object Replacement**, we focus on replacing the object in the action-object pair. For instance, revisiting the previous example, we replace the object *keyboard* with *bread*. Here, we prefer to use a set of token-based MLMs (Devlin et al., 2019; Lan et al., 2020; Liu et al., 2019). To further enhance the quality of the foils, we opt for an ensembling approach in the object replacement test. More details are given in Appendix C.5.

### 3.6 SPATIAL RELATIONS

The **Spatial Relations** test focuses on the ability of models to distinguish different spatial and spatio-temporal relations related to the actions carried out in a video (e.g. moving an object *'over'*, or *'towards'* another object). It is similar to the relation task introduced in Parcalabescu et al. (2022), with the notable difference that the model must use temporal information to accomplish the task. We create the foils starting from the Something-Something V2 validation set (Goyal et al., 2017a) which contains 174 pre-defined actions with everyday objects. To create a candidate foil, we replace the spatial preposition with an in-distribution alternative, drawn from the set of spatial prepositions in the validation set. We rank the candidate foils by scoring their plausibility using T5 (Raffel et al., 2020) and select the top 10 best-scoring foils. We then use the GRUEN pretrained model (Zhu & Bhat, 2020) to score the foils for grammaticality, keeping foils with scores higher than $0.6$. We filter caption-foil pairs with an NLI model, keeping only foils classified as neutral or contradiction with respect to the caption. Finally, we smooth out the foil distribution to match the original validation distribution. This mitigates distribution biases arising in the foil generation process, which could be exploited by the tested models. Full details are provided in Appendix C.6.

### 3.7 HUMAN VALIDATION

A central requirement for VILMA is to ensure validity, that is, humans should agree that captions are true of the videos, while foils are not. We validated the entire VILMA dataset in two separate stages. For the simpler proficiency tests, we manually checked every video-caption-foil sample, retaining only those in which the foil was unambiguously false with respect to the input video. This resulted

in the removal of 1278 (15.11%) of samples in the proficiency tests. The main tests were validated independently, in a study conducted on AMTurk. Each sample was evaluated by three independent annotators, who were asked to judge which text (caption or foil), if any, was true of the video. See Appendix B for details on method, annotators and qualification tasks. We retained only samples for which at least two out of three independent annotators judged only the caption as true of the video, resulting in a final set of 5177 (61.19%) of the initial set. See Appendix B.1 for details by task.

## 4 EXPERIMENTS

### 4.1 PRETRAINED MODELS

We analyse the performance of 12 architecturally diverse, state-of-the-art VidLMs: ClipBERT (Lei et al., 2021), UniVL (Luo et al., 2020), VideoCLIP (Xu et al., 2021), FiT (Bain et al., 2021), CLIP4Clip (Luo et al., 2022), VIOLET (Fu et al., 2021), X-CLIP (Ma et al., 2022), MCQ (Ge et al., 2022), Singularity (Lei et al., 2022), UniPerceiver (Zhu et al., 2022), Merlot Reserve (Zellers et al., 2021), VindLU (Cheng et al., 2023), InternVideo (Wang et al., 2022c), mPLUG-2 (Xu et al., 2023), Otter (Li et al., 2023b) and Video-LLaMA (Zhang et al., 2023). The models were trained on different tasks and data. We also benchmark two commonly used ILMs: CLIP (Radford et al., 2021) and BLIP-2 (Li et al., 2023d), alongside two unimodal baselines: GPT-2 (Radford et al., 2019) and OPT (Zhang et al., 2022). See Appendix A for a detailed overview of models.

### 4.2 EVALUATION METRICS

For our evaluation, we rely on the straightforward yet informative metric of **pairwise ranking accuracy** denoted as $acc_r$. This metric essentially measures the proportion of samples where the video-caption matching score surpasses the video-foil matching score. The primary choice of **pairwise accuracy** allows us to directly compare all 12 VidLMs, including VidLMs that were pretrained using both VTC and NLG objectives. We report $acc_r$ scores for both the main tests (T) and their respective proficiency tasks (P). Additionally, we introduce a more strict combined score (P+T), wherein a model's success on the main test is only deemed correct if it also succeeds on its proficiency test. Finally, we take the average of combined scores (P+T) among each task to provide a summary score for each model.

### 4.3 RESULTS AND ANALYSIS

Table 2 offers a concise overview of our results. For a more in-depth analysis, including per-subtest outcomes, we refer readers to the Appendix C. ' .

**Unimodal Results.** The unimodal baselines perform close to the random baseline in Counting and Change of State, but not in the remaining tests. In Rare Actions, this outcome is expected given that the captions inherently describe *less likely events*. Similarly, within the proficiency test for the Change of State, we introduce the foiling of low-frequency nouns (e.g., hyponyms) with high-frequency ones (e.g., hypernyms), which inadvertently biases the model towards favouring the foils. In contrast, unimodal models exhibit a notably superior performance compared to the random baseline in Situation Awareness and Spatial Relations. This can be partially attributed to *plausibility biases* (Madhyastha et al., 2019; Parcalabescu et al., 2022) introduced during foil generation. The shared linguistic context between the caption and foil constrains the selection of foiling actions/relations, often leading to the introduction of unlikely or unnatural alternatives.

**Image-Language Model Results.** Much like the unimodal baselines, the performance of ILMs in the Counting and Change of State tasks is close to random. However, we note that ILMs exhibit proficiency in detecting objects and capturing semantics, as shown in the proficiency test results for Rare Actions and Counting, where the former requires object detection capabilities, and the latter hinges on precise action recognition. In several tasks, ILMs even outperform their VidLM counterparts. For instance, BLIP2 is the best-performing model in Situation Awareness, while in the Rare Actions task, CLIP performs better than all the other models excluding VindLU.

**Video-Language Model Results.** In the majority of tasks, VidLMs deliver performance levels that closely resemble those of ILMs. This observation raises a critical point: the temporal reasoning capa-

Table 2: Pairwise ranking accuracy ($acc_r$) performance of 12 Video-Language Models on the VILMA benchmark on the proficiency (**P**), main (**T**), and combined (**P+T**) tasks. In the combined task **P+T**, a success in **T** only counts if **P** is also successful. The final column **Avg.** is the taskwise average of combined scores **P+T** among each task. Best (second-best) model per metric are marked in boldface (underlined). More in-depth analysis of the experiments are given in Appendix C.

| Model | Action Counting | | | Situ. Awareness | | | Change of State | | | Rare Actions | | | Spatial Relations | | | Avg. |
|---|---|---|---|---|---|---|---|---|---|---|---|---|---|---|---|---|
| | P | T | P+T | P | T | P+T | P | T | P+T | P | T | P+T | P | T | P+T | P+T |
| Random | 50.0 | 50.0 | 25.0 | 50.0 | 38.0 | 19.0 | 50.0 | 50.0 | 25.0 | 50.0 | 50.0 | 25.0 | 50.0 | 50.0 | 25.0 | 23.8 |
| GPT-2[†] | 50.3 | 53.3 | 27.6 | 44.6 | 66.6 | 31.7 | 18.0 | 52.4 | 10.8 | 58.4 | 25.9 | 17.7 | 49.1 | 72.8 | 43.0 | 26.2 |
| OPT[†] | 56.2 | 54.6 | 31.0 | 51.7 | 71.4 | 38.7 | 23.1 | 48.0 | 12.9 | 59.0 | 23.9 | 14.9 | 59.0 | 84.7 | 55.7 | 30.6 |
| CLIP[‡] | 90.5 | 50.9 | 46.2 | 71.0 | 45.6 | 33.7 | 93.0 | 55.2 | 52.2 | 92.7 | 93.9 | 87.8 | 78.6 | 58.3 | 44.8 | 52.9 |
| BLIP2[‡] | 80.9 | 54.5 | 43.7 | 73.4 | 75.4 | 55.8 | 74.5 | 52.1 | 38.1 | 93.8 | 74.5 | 70.5 | 91.1 | 86.0 | 79.4 | 57.5 |
| ClipBERT | 56.4 | 49.6 | 28.0 | 54.1 | 57.0 | 31.9 | 63.7 | 50.0 | 33.5 | 43.5 | 40.7 | 17.7 | 39.7 | 39.8 | 14.1 | 25.0 |
| UniVL | 73.4 | 43.6 | 32.2 | 52.9 | 46.7 | 24.1 | 81.3 | 54.3 | 43.0 | 77.5 | 78.0 | 59.9 | 62.5 | 51.7 | 33.2 | 38.5 |
| VideoCLIP | 79.1 | 46.4 | 36.5 | 61.7 | 40.4 | 24.9 | 49.8 | 50.8 | 25.9 | 84.0 | 77.8 | 67.5 | 67.9 | 54.7 | 39.7 | 38.9 |
| FiT | 83.9 | 52.4 | 44.6 | 69.8 | 40.1 | 29.2 | 93.0 | 52.1 | 47.8 | 89.7 | 89.4 | 80.7 | 70.5 | 51.9 | 38.7 | 48.2 |
| CLIP4Clip | 91.2 | 52.3 | 48.0 | 73.9 | 49.1 | 37.7 | 94.8 | 54.1 | 52.1 | 83.0 | 94.1 | 78.7 | 79.8 | 56.7 | 44.2 | 52.1 |
| VIOLET | 79.6 | 50.6 | 36.5 | 70.3 | 44.5 | 32.5 | 88.2 | 54.6 | 49.1 | 87.1 | 86.6 | 74.6 | 73.3 | 50.4 | 38.7 | 46.3 |
| X-CLIP | 84.1 | 55.1 | 46.4 | 63.6 | 44.9 | 31.1 | 85.7 | 52.7 | 46.0 | 83.9 | 85.7 | 72.3 | 74.8 | 56.2 | 43.5 | 47.8 |
| MCQ | 81.4 | 50.4 | 41.5 | 67.1 | 37.1 | 26.3 | 90.3 | 50.3 | 45.3 | 91.3 | 88.7 | 82.3 | 79.4 | 48.9 | 39.4 | 47.0 |
| Singularity | 79.6 | 51.1 | 41.5 | 68.8 | 40.9 | 30.2 | 92.8 | 54.6 | 50.3 | 92.7 | 88.4 | 83.1 | 80.7 | 46.8 | 38.9 | 48.8 |
| UniPerceiver | 50.6 | 46.4 | 23.0 | 51.5 | 42.2 | 21.2 | 67.5 | 46.1 | 29.1 | 58.2 | 58.8 | 34.7 | 45.5 | 48.0 | 20.1 | 25.6 |
| Merlot Reserve | 84.2 | 56.0 | 46.9 | 70.6 | 35.7 | 25.4 | 93.4 | 53.6 | 50.4 | 83.8 | 90.6 | 77.6 | 63.1 | 41.9 | 29.2 | 45.9 |
| VindLU | 84.5 | 51.2 | 43.5 | 70.6 | 41.6 | 31.3 | 85.4 | 52.6 | 45.6 | 94.2 | 93.1 | 88.0 | 83.2 | 45.6 | 39.4 | 49.5 |
| InternVideo | 90.2 | 54.3 | 48.7 | 71.6 | 41.1 | 29.5 | 95.6 | 57.7 | 55.1 | 95.6 | 96.7 | 92.7 | 76.6 | 59.8 | 45.3 | 54.2 |
| mPLUG-2 | 57.7 | 49.7 | 27.7 | 49.6 | 37.4 | 21.5 | 39.5 | 47.7 | 20.8 | 50.8 | 47.0 | 24.0 | 46.6 | 48.1 | 26.5 | 24.1 |
| Otter | 59.4 | 52.7 | 30.7 | 58.8 | 51.0 | 29.3 | 65.7 | 53.0 | 34.3 | 56.1 | 58.8 | 35.6 | 62.9 | 71.3 | 47.6 | 35.5 |
| Video-LLaMA | 84.6 | 56.3 | 47.3 | 78.2 | 67.0 | 54.0 | 81.4 | 59.0 | 46.8 | 78.7 | 71.0 | 58.6 | 88.6 | 88.8 | 79.6 | 57.3 |

bilities of current VidLMs are evidently far from adequate. Remarkably, in the Counting, Situation Awareness, and Change of State tests, many VidLMs do not show a notably higher performance than the random baseline. Our findings highlight the urgent need for the community to prioritise and enhance the temporal reasoning abilities of these models.

**Proficiency Results.** The results reveal that both ILMs and VidLMs tend to consistently perform better in the simpler proficiency test, with few exceptions. These tests provide valuable insights by enabling a more robust evaluation of models. An intriguing insight emerges from the evaluation of models in the **combined setting**, where a striking performance drop occurs. This suggests that in a substantial number of cases, when models predict correct answers in the main tasks, they do so by chance or due to reliance on spurious features, rather than due to a robust understanding of the input.

## 5 CONCLUSION

We introduced VILMA, a video-language foiling benchmark, which probes the capabilities of pretrained VidLMs where both commonsense and temporal reasoning take centre-stage. We have conducted a comprehensive evaluation and comparison of numerous VidLMs as well as ILMs and text-only LMs on our benchmark. Our experiments show that, as far as visually grounded temporal reasoning abilities are concerned, VidLMs do not differ substantially from ILMs. To further refine our benchmark, we introduced proficiency tests, which not only enhance granularity but also provide deeper insights into the models' aptitude. Strikingly, our proficiency task results reveal that a considerable portion of correct predictions appears to be accidental rather than indicative of robust understanding. This highlights that current VidLMs struggle with the intricacies of temporal reasoning. It also underlines the importance of benchmarks like VILMA to identify weaknesses of current VidLMs that need improvement.

## ACKNOWLEDGMENTS

IC has received funding from the European Union's Horizon 2020 research and innovation programme under the Marie Skłodowska-Curie grant agreement No 838188. AG and MC are supported by the European Union's Horizon 2020 research and innovation Programme under the Marie Skłodowska-Curie grant agreement No 860621.

This publication is based upon work from the COST Action Multi3Generation CA18231, supported by COST (European Cooperation in Science and Technology). It was supported in part by AI Fellowships to IK and EA provided by the KUIS AI Center. MC and AG are supported by Marie Skłodowska-Curie grant agreement No 860621 to the NL4XAI (*Natural Language for Explainable AI*) under the European Union's Horizon 2020 research and innovation programme. AP was supported by the European Commission (Grant 951911) under the H2020 Programme ICT-48-2020, and by the FAIR project, funded by the Italian Ministry of University and Research under the NextGenerationEU program.

## REPRODUCIBILITY STATEMENT

We share[5] the code and documentation to replicate our experiments, including the tools to generate both proficiency and main tests. The models utilised in our assessment were sourced from the checkpoints provided by their respective authors or projects. We release our code under the same licensing terms, ensuring transparency and reproducibility.

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

APPENDIX

**Appendix A** provides further descriptions of the models that we include in the benchmark together with implementation details. **Appendix B** presents a detailed report of our data validation process, annotator selection criterion, annotation statistics, inter-annotator agreements, bias check, and annotation costs. **Appendix C** gives additional details of each test (e.g. data sources, foiling methods), and presents in-depth analysis of the evaluated models on our tests. Finally, in **Appendix D**, we present additional analyses detailing how model performance varies in response to various factors.

## A    PRETRAINED MODELS

Here, we describe the models used in this benchmark. Next to the pretrained video-language models (§A.3), we also experimented with pretrained unimodal models (i.e. text-only LMs) and image-language models (§A.1 and §A.2).

### A.1    UNIMODAL MODELS

We test a couple of decoder-only or encoder-decoder LMs on the benchmark. These models are GPT-2 (Radford et al., 2019), OPT (Zhang et al., 2022), T5 (Raffel et al., 2020) and BART (Lewis et al., 2020). Similar to VALSE, we calculate the perplexity values for both caption and foil, and select the text input with smaller perplexity score. For our experiments with GPT-2 and OPT, we use GPT-2[6] with 124M parameters and OPT-6.7B[7].

### A.2    IMAGE-LANGUAGE MODELS

We also conducted experiments involving two prominent Image-Language Models: CLIP (Radford et al., 2021) and BLIP-2 (Li et al., 2023d). CLIP employs a dual-encoder architecture, with a contrastive loss as objective to facilitate the training of image-caption pairs. On the other hand, BLIP-2 represents a subsequent advancement of BLIP (Li et al., 2022b), harnessing the potential of frozen pretrained image encoders and large language models to bolster the vision-language learning process. For CLIP and BLIP-2 experiments, we use the largest version of CLIP[8] and BLIP-2[9] with OPT-6.7B.

### A.3    VIDEO-LANGUAGE MODELS

In this section, we share the details of the pretrained video-language models previously listed in §4.1. Table 3 gives a systematic overview of these models based on their architectures and pretraining procedures, which are categorised in Section 2.1. §A.4 shares the implementation details of these models.

**ClipBERT** (Lei et al., 2021) uses BERT (Devlin et al., 2019) as text encoder and ResNet-50 (He et al., 2016) as video encoder. Unlike others, it is pretrained using solely images (Lin et al., 2014; Krishna et al., 2016). Moreover, ClipBERT is unable to learn temporal ordering: the video-text similarity score is the average frame-text similarity score.

**UniVL** (Luo et al., 2020) is a two-stream encoder-decoder model. A pretrained BERT encodes the textual input, whereas visual features are extracted via S3D and processed by a transformer encoder. Modalities are fused via a cross-encoder. UniVL is pretrained on HowTo100M and, unlike many VidLMs, it is also trained on a generative task.

**VideoCLIP** (Xu et al., 2021) uses BERT as text encoder and S3D (Xie et al., 2018) as video encoder. VideoCLIP is pretrained on HowTo100M. Like ClipBERT, it uses mean pooling to fuse modalities.

---

[6]https://huggingface.co/gpt2

[7]https://huggingface.co/facebook/opt-6.7b

[8]https://huggingface.co/openai/clip-vit-large-patch14

[9]https://huggingface.co/Salesforce/blip2-opt-6.7b

Table 3: A systemic comparison of the VidLMs included in VɪLMA. We categorise these methods based on their model architectures and pretraining procedures. The last column contains the task-wise average **P+T** scores. The terms and acronyms are defined in the table footer.

| Method | Model | | | Pretraining | | | Score |
|---|---|---|---|---|---|---|---|
| | Temporal Modeling | Multimodal Fusion | Pretraining Objectives | Dataset | Size | Modality | |
| ClipBert | Mean Pooling | BERT | MLM+VTM | COCO+VG | 0.2M | I | 25.0 |
| UniVL | Temp. Att. | 2-layer TR | VTC+VTM+MLM+MFM+NLG | HT | 136M | V | 38.5 |
| VideoCLIP | 1D-Conv+TR | - | VTC | HT | 136M | V | 38.9 |
| FiT | Temp. Att. | - | VTC | C5M | 5M | I+V | 48.2 |
| CLIP4Clip | Late TR | - | VTC | CLIP | 400M | I | 52.1 |
| VIOLET | Window Att. | BERT | VTC+VTM+MLM+MVM | YT+C5M | 185M | I+V | 46.3 |
| X-CLIP | Temp. Att. | - | VTC | CLIP | 400M | I | 47.8 |
| MCQ | Temp. Att. | 12-layer TR | MLM+VTC | C5M | 5M | I+V | 47.0 |
| Singularity | Late Temp. Att. | 3-layer TR | VTC+VTM+MLM | C17M | 17M | I+V | 48.8 |
| UniPerceiver | Temp. Att. | BERT | VTC | Custom | 45M | I+V | 25.6 |
| Merlot Reserve | Temp. Att. | 24-layer TR. | MLM+MAM+FOM | YT | 20M | V+A | 45.9 |
| VindLU | Temp. Att. | 3-layer TR | VTC+VTM+MLM | C25M | 25M | I+V | 49.5 |
| InternVideo | Temp. Att. | Decoder TR | VTC+NLG+MVM | Custom | 12M | I+V | 54.2 |
| mPLUG-2 | Temp. Att. | 6-layer TR | VTC+MLM+NLG | C17M | 17M | I+V | 24.1 |
| Otter | - | MPT | NLG | Custom | 2.8M | I+V | 35.5 |
| Video-LLaMA | Late TR | Vicuna/LLaMA | NLG | Custom | 5M | A+I+V | 57.3 |

**TR**: Transformer; **Late**: Late fusion; **Att**: Attention. **V**: Video; **I**: Image; **A**: Audio. **VTC**: Video-text contrastive; **VTM**: Video-text matching; **MLM**: Masked language modeling; **MVM**: Masked video modeling; **MFM**: Masked frame modeling; **NLG**: Natural language generation. **MPT**: MosaicML Pretrained Transformer (Team et al., 2023). **HT**: HowTo100M (Miech et al., 2019); **C5M**: CC3M (Sharma et al., 2018) and WebVid-2 (Bain et al., 2021); **C17M**: Combination of C5M, COCO, VG, SBU captions (Ordonez et al., 2011) and CC12M (Changpinyo et al., 2021) datasets. **YT**: YT-Temporal-1B (Zellers et al., 2021); **COCO**: (Lin et al., 2014), **VG**: Visual Genome (Krishna et al., 2016); **Custom**: A custom dataset, please see the original work.

**FiT** (Bain et al., 2021) encodes text using BERT like many others. As video encoder, TimeSFormer (Bertasius et al., 2021) is preferred. FiT is pretrained on both images (CC3M) and videos (W2). It creates a shared video-text space via contrastive learning. The authors also collected the W2 dataset.

**CLIP4Clip** (Luo et al., 2022) model seeks to utilise the CLIP (Radford et al., 2021) model's knowledge for end-to-end video-language retrieval. The authors carry out empirical research to answer significant issues, such as whether image features are sufficient for video-text retrieval, how post-pretraining using CLIP affects a large video-text dataset, how to model temporal dependency between video frames, and how hyperparameters affect video-text retrieval.

**VIOLET** (Fu et al., 2021) is a dual-stream encoder-only architecture. The textual module is initialised from pretrained BERT-base. Video frames are uniformly sampled and processed by a Video Swin Transformer (Liu et al., 2022) encoder. Spatial and temporal dimensions of the video inputs are modelled by positional embeddings considering both spatial and temporal ordering. VIOLET is pretrained on videos (YT-Temporal, WebVid) and images (CC3M). All modules are tuned in training.results

**X-CLIP** (Ma et al., 2022) is a video-text retrieval model that offers a new approach to address the challenge of similarity aggregation. By employing a multi-grained contrastive mechanism, the model encodes sentences and videos into coarse-grained and fine-grained representations, facilitating contrasts across different levels of granularity. Moreover, the model introduces the Attention Over Similarity Matrix (AOSM) module, enabling it to focus on essential frames and words while reducing the impact of irrelevant ones during retrieval.

**MCQ** (Ge et al., 2022) introduced a novel pretraining task, Multiple Choice Questions (MCQ) for the VidLMs based on a dual-encoder mechanism. They used a parametric module called BridgeFormer, which connects local features from VideoFormer (Dosovitskiy et al., 2020) and TextFormer (Sanh et al., 2019) to answer multiple-choice questions via contrastive learning objective. It enhances

semantic associations between video-text representations and improves fine-grained semantic associations between two modalities. Additionally, it maintains high efficiency for retrieval and the BridgeFormer can be removed for downstream tasks.

**Singularity** (Lei et al., 2022) showed the effectiveness of single-frame training in the context of VidL tasks, such as video question answering and text-to-video retrieval, by incorporating a vision encoder (Dosovitskiy et al., 2020), a language encoder (Devlin et al., 2019), and a multi-modal encoder with cross-attention fusion mechanism. On the other hand, they have implemented a new benchmark to overcome focusing on models temporal learning abilities. This contribution brings to light a significant static appearance bias prevalent in current video-and-language datasets.

**UniPerceiver** (Zhu et al., 2022) is primarily concerned with pretraining a single framework for general perception tasks. The model is designed to handle zero-shot and few-shot learning situations. It integrates the capabilities of transformers with neural perceptrons to enable successful learning using a variety of modalities, including texts, audio, and images. UniPerceiver does this through the use of a common encoder-decoder structure, which allows it to capitalize on the correlations between multiple modalities throughout pretraining.

**Merlot Reserve** (Zellers et al., 2021) improves video comprehension by combining audio, subtitles, and video frames. The model learns by substituting bits of text and audio with a MASK token and selecting the proper masked-out piece. MERLOT Reserve's training aim beats alternatives, and the model obtains outstanding scores when used for challenges like Visual Commonsense Reasoning (Zellers et al., 2019), TVQA (Lei et al., 2018), and Kinetics-600 (Carreira et al., 2018).

**VindLU** (Cheng et al., 2023) followed a comprehensive approach for enhancing VidL pretraining to fine the most effective VidL framework design. The methodology begins by employing image (Bao et al., 2021) and text (Devlin et al., 2019) encoders, trained on video and caption pairs through a visual-text contrastive objective. Subsequently, the framework progressively incorporates additional components while analyzing the significance of each one. The final recipe encompasses six steps, which involve the inclusion of temporal attention, integration of a multimodal fusion encoder, adoption of masked modeling pretraining objectives, joint training on images and videos, utilization of additional frames both in fine-tuning and inference stages, model-parameter and data scaling. These steps collectively contribute to an effective VidL pretraining process, facilitating improved performance and understanding in multimodal video question answering tasks.

**InternVideo** (Wang et al., 2022c) proposes a novel approach addressing limitations in existing vision foundation models by focusing on video-level understanding tasks. InternVideo combines generative and discriminative self-supervised video learning, utilizing masked video modeling and video-language contrastive learning as pretraining objectives. By coordinating representations from these frameworks, InternVideo significantly improves performance across diverse video applications. It achieves state-of-the-art results on 39 video datasets, including tasks like action recognition/detection, video-language alignment, and open-world video applications, attaining 91.1% and 77.2% top-1 accuracy on Kinetics-400 (Carreira & Zisserman, 2017) and Something-Something V2 (Goyal et al., 2017a) benchmarks respectively.

**mPLUG-2** (Xu et al., 2023) is a foundational model, which aims to unify several modalities including language, image and video, similar to UniPerceiver (Zhu et al., 2022). To do so, they implement cross-modal transformer layers, which produce visually-aware textual features and textually-aware visual features. Afterwards, a fusion module (Li et al., 2022a) jointly processes these features. Finally, a text decoder is employed to adapt the method to the generative tasks. mPLUG-2 processes 14M image-text pairs (COCO+VG+SBU+CC3M) and 2.5M video-text pairs (WebVid2M) during its pretraining phase.

**Otter** (Li et al., 2023b) is a multimodal model based on OpenFlamingo (Awadalla et al., 2023) framework and specialized in multi-modal in-context instruction tuning, using the MIMIC-IT (Li et al., 2023a) dataset to enhance its ability to process and respond to instructions both for video and multiple image inputs. After finetuned on MIMIC-IT with 2.8 million multimodal instruction-response pairs, Otter shows improved instruction-following abilities compared to OpenFlamingo.

**Video-LLaMA** (Zhang et al., 2023) is a conversational VidLM , which can follow the given instructions. Its foundations are based on BLIP2 (Li et al., 2023d): two separate query-formers (Q-Former) process video and audio modalities to produce their query embeddings, which are prepended into a

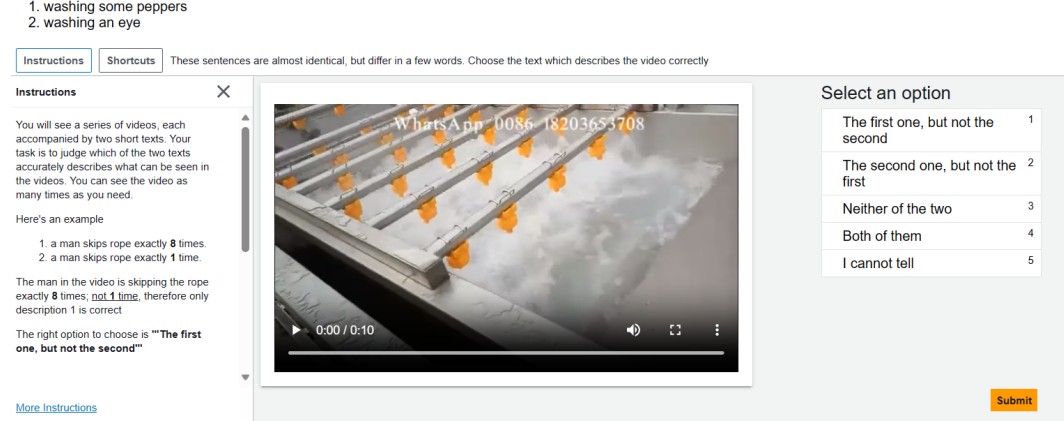

Figure 2: Form used in the human validation. The general instructions on the left-hand side are always visible to the annotator.

frozen language model as its prefix. Vicuna (Zheng et al., 2023) and LLaMA models (Touvron et al., 2023) are used as language models. During the pretraining phase, WebVid2M and a subset of CC3M are used as data resources. Later, it is fine-tuned on the Video-Chat instructions dataset (Li et al., 2023e). We used the Video-LLaMA model with LLaMA-7B in our experiments.

### A.4 IMPLEMENTATION DETAILS

We try to use each model *as-is* based on the provided official implementations in a zero-shot setting. We directly use Huggingface implementations (Wolf et al., 2019) of GPT-2, OPT, CLIP, BLIP2 and X-CLIP. The majority of VidLMs sample a model-specific number of frames $K$ to construct video input. Specifically, X-CLIP, InternVideo and Video-LLaMA utilise $K = 8$, while ClipBERT operates with $K = 16$. Meanwhile, the other tested models maintain a value of $K = 4$. VideoCLIP, CLIP4Clip, and UniVL process the entire video using a S3D video encoder (Xie et al., 2018). The distinctive methodology employed by the Merlot Reserve model involves the selection of a time interval, wherein the input video is systematically partitioned into segments according to this predetermined temporal span. Subsequently, the model meticulously captures the middle frame within each interval. We set time interval as 5 seconds as used in Merlot Reserve. In cases where the video duration falls below the specified 5-second interval, we captured the central frame. To calculate video-caption match scores for ILMs, we perform mean pooling over the image-caption match scores obtained using multiple frames, setting $K = 8$. We run experiments on single Tesla T4, Quadro P4000 or V100 GPUs using half precision.

## B VILMA VALIDATION

We run a thorough human validation of VILMA, validating both the main test cases (detailed description in §B.1) and the proficiency tests (described in detail in § B.2). We report the total number of valid cases for all the tests in Table 6.

### B.1 AMAZON MECHANICAL TURK ANNOTATION AND EVALUATION

**Setup** We ran a human validation of each test and subtest in VILMA. Annotators were shown an instance composed of a video and two descriptions, namely a caption and a foil as shown in Figure 2. The annotators received the following general instructions:

> *You will see a series of videos, each accompanied by two short texts. Your task is to judge which of the two texts accurately describes what can be seen in the videos. You can see the video as many times as you need.*

For each case, along with the general instructions, the video, and the two descriptions, the annotator was instructed as follows: *"These sentences are almost identical, but differ in a few words highlighted in boldface. Choose the text which describes the video correctly".* Following this, five possible answers were given: (1) The first one, but not the second, (2) The second one, but not the first, (3) Neither of the two, (4) Both of them, (5) I cannot tell. The order of the two descriptions (caption and foil) were randomised so that the caption appeared in the first position $50\%$ of the time. We collect three annotations for each sample.

**Annotator Selection**    We used the proficiency test present in each test in VɪLMA as a qualification task to recruit qualified annotators for our validation. As mentioned in Section 3.1, the proficiency test in VɪLMA, can be considered as a preliminary criterion for each test and therefore, it is a natural selection strategy to identify potential good annotators. Note that, apart from their use for annotator selection, proficiency tests were also independently validated (see § B.2.

For each test, we chose 1 subtest and we asked the annotators to assess the proficiency test annotations, by using the same setup shown in Figure 2. We use 5 proficiency tests in total (i.e. *Change-State-Reverse, Counting-Easy, Rare Action-Action Replacement, Spatial Relations-Prepositions, Situation Awareness-Action Replacement*), with an additional sanity check consisting of a proficiency test (i.e. *Spatial Relations-Prepositions*) where all the videos and the caption-foil pairs were mismatched (and thus the annotators were always expected to answer (3) "Neither of the two"). The whole setup accounts for a total of 4977 instances for which we collected 3 annotations each.

Moreover, we asked two expert annotators to manually annotate a batch of 10 randomly sampled instances per proficiency test. The purpose of this manual annotation was two-fold: (i) produce gold annotations to use as further filtering in the recruitment process, (ii) identify baseline accuracy scores for the proficiency tests. We observed an average accuracy between the expert annotators of $80\%$.

We recruited annotators who had an approval rating of $90\%$ or higher on Amazon Mechanical Turk and had correctly identified the caption in the proficiency test at least $90\%$ of the time (higher than the observed baseline accuracy of $80\%$) with a minimum of 7 instances annotated. Based on this, we recruited a total of 101 annotators who finally participated in the VɪLMA test validation.

**Results**    In Table 4 we show the statistics relevant to the human validation of our tests. For each subtest, we report the number of valid instances, namely instances where 2 out of 3 annotators chose the caption but not the foil, as well as the number of unanimous annotations, namely when $3/3$ annotators chose the caption. The proportion of valid instances can vary according to the test, but overall we observe that the $70\%$ of the total number of instances in VɪLMA are judged valid by humans, and thus they can be considered high-quality caption-foil pairs.

**Annotator Agreement**    Table 4 also reports the inter-annotator agreement between annotators in the validation, using Krippendorff's $\alpha$ Krippendorff (1989) computed overall and over the valid instances. The agreement for the valid instances is higher and ranges from $0.1$ to $0.4$. The low to medium agreement is due to two main reasons: first, we compute the agreement over the whole pool of annotators, who may have annotated quite different numbers of samples (ranging from 7 to $103$); second, during the validation task, annotators had to choose one out of $5$ responses. This is different from VɪLMA, where all tests are binary tasks.

**Bias Check**    Although distributional biases between foils and caption were taken into account in the construction of VɪLMA (as described in §3), after the human validation such biases may be reintroduced. To check for this, we compare the word frequency distributions between the original tests and the human-validated ones. We report the Jensen-Shannon divergence (JS) of the two distributions in Table 4, while caption foil distributions for each test are reported in Figures 3-10.

The Jensen-Shannon Divergence is defined as follows:

$$JS(f \parallel c) = \sqrt{\frac{KL(f \parallel m) + KL(c \parallel m)}{2}}$$

where $f$ is the normalized word frequency for foils, $c$ the normalized word frequency for captions, $m$ is the point-wise mean of $f$ and $c$, and $KL$ is the Kullback-Leibler divergence.

Table 4: Manual validation results for each test in VɪLMA. *#Inst.*: number of instances related to a linguistic phenomenon. *#Valid (%)*: number (percent) of cases for which at least 2 out of 3 annotators chose the caption; *#Unan. (%)*: number (percent) of cases for which all annotators chose the caption; *#Lex.It.*: number of phrases or lexical items in the vocabulary that differ between foils and captions; *JS*: Jensen-Shannon divergence between foil-caption distributions for all instances in the whole subtest; *JS Val.*: Jensen-Shannon divergence between foil-caption distribution for the valid instances of the subtest, after sub-sampling; $\alpha$: Krippendorff's $\alpha$ coefficient computed over all the instances; $\alpha$ *valid*: Krippendorff's $\alpha$ coefficient computed over the *Valid* instances.

| Test | Subtest | #Inst. | #Valid (%) | #Unan. (%) | #Lex.it. | JS | JS Val. | $\alpha$ | $\alpha$ Valid |
|---|---|---|---|---|---|---|---|---|---|
| **Change of State** | *Action* | 624 | 466(74.68) | 201(32.21) | 50 | 0.311 | 0.301 | 0.183 | 0.303 |
| | *Pre-State* | 624 | 286(45.83) | 80(12.82) | 2 | 0.146 | 0.129 | 0.017 | 0.106 |
| | *Post-State* | 624 | 383(61.38) | 111(17.79) | 1 | 0.146 | 0.151 | 0.059 | 0.145 |
| | *Reverse* | 624 | 342(54.81) | 109(17.47) | 48 | 0.148 | 0.138 | 0.070 | 0.183 |
| **Action Counting** | *Easy* | 959 | 774(80.71) | 428(44.63) | 0 | 0.085 | 0.084 | 0.340 | 0.453 |
| | *Difficult* | 895 | 682(76.20) | 274(30.61) | 2 | 0.077 | 0.076 | 0.148 | 0.251 |
| **Rare Actions** | *Action Replacement* | 978 | 781(79.86) | 353(36.09) | 9 | 0.485 | 0.479 | 0.222 | 0.333 |
| | *Object Replacement* | 972 | 739(76.03) | 307(31.58) | 6 | 0.450 | 0.442 | 0.186 | 0.299 |
| **Spatial Relations** | *Prepositions* | 708 | 436(61.58) | 132(18.64) | 2 | 0.030 | 0.039 | 0.067 | 0.167 |
| **Situation Awareness** | *Action Replacement* | 1000 | 838(83.80) | 377(37.70) | 60 | 0.176 | 0.175 | 0.224 | 0.313 |
| | *Actor Swapping* | 452 | 207(45.80) | 61(13.50) | 5 | 0.025 | 0.022 | 0.026 | 0.204 |
| **Overall** | | 8460 | 5934(70.14) | 2433(28.76) | | | | | |

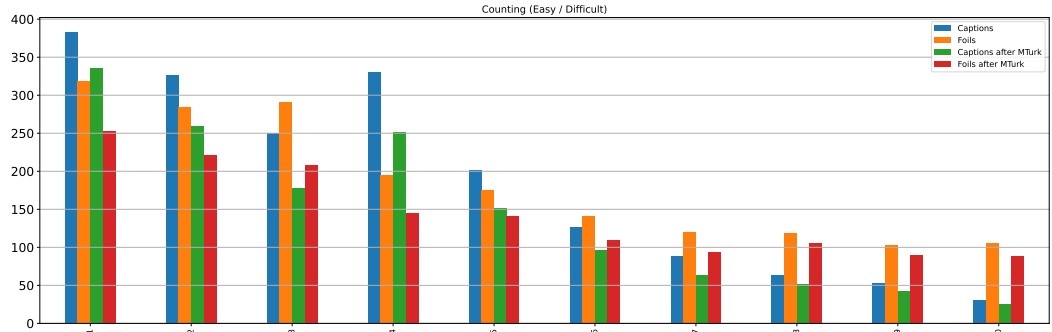

Figure 3: Caption and foil distribution of Action Counting test, before and after Amazon Mechanical Turk validation process.

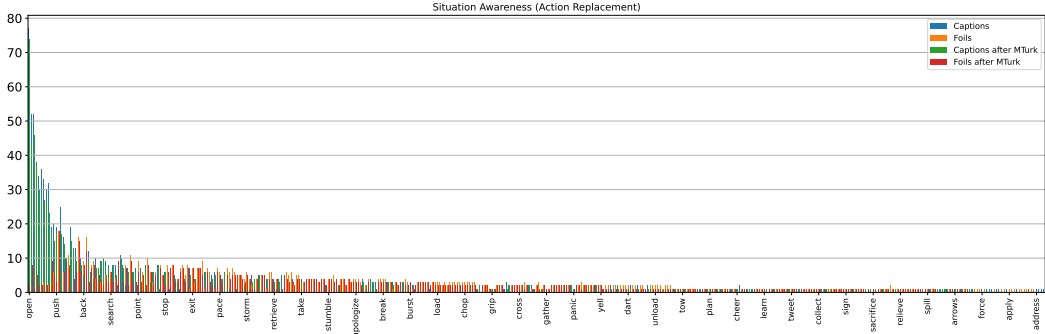

Figure 4: Caption and foil distribution of Situation Awareness main test, before and after Amazon Mechanical Turk validation process.

As shown in Table 4, the JS marginally changes after the human validation. Moreover, we observe minimal lexical differences (see the *#Lex. it.* column) in the vocabulary distributions. This suggests that biases are not significantly present in the validated data, that is, there are few if any lexical cues that could be used by a model to spuriously identify a foil versus a caption in the tests.

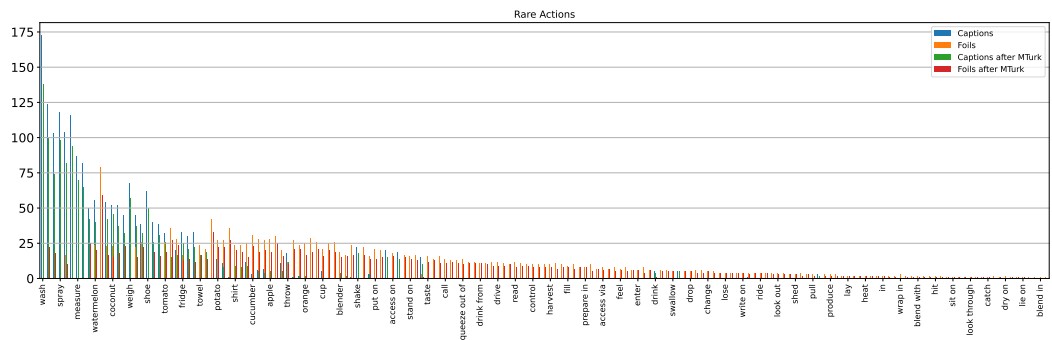

Figure 5: Caption and foil distribution of Rare Actions test, before and after Amazon Mechanical Turk validation process.

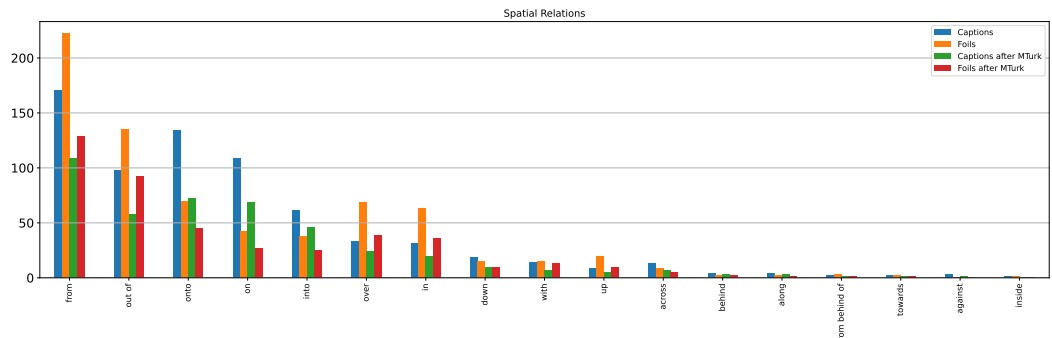

Figure 6: Caption and foil distribution of Spatial Relations test, before and after Amazon Mechanical Turk validation process.

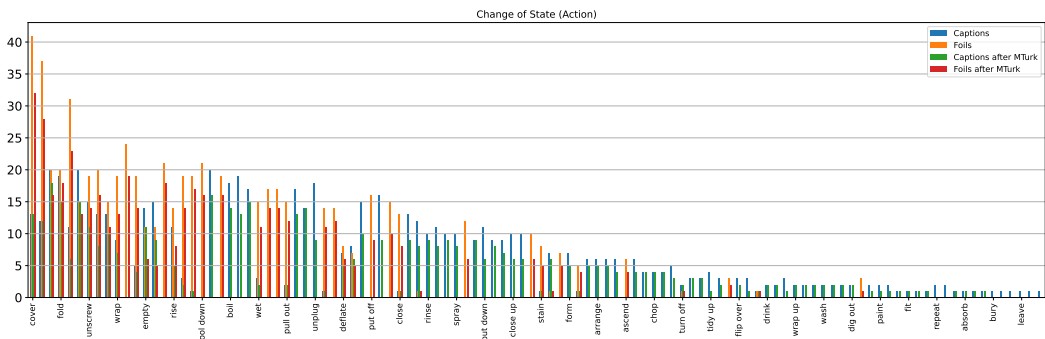

Figure 7: Caption and foil distribution of Change of State - Actions test, before and after Amazon Mechanical Turk validation process.

**Annotation Costs**    Annotators were paid $0.05 per item (i.e. per HIT on Mechanical Turk). The whole validation – including the qualification task – cost around $2100.

## B.2    PROFICIENCY TEST VALIDATION

**Setup**    In contrast to the main tests, we opt for internal validation of the proficiency tests in VILMA. This decision stems from the lower complexity of the proficiency tests, both in its creation process and in its definition.

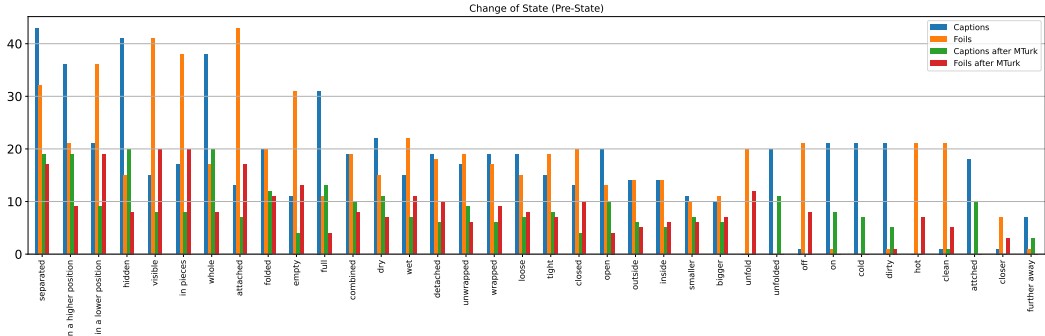

Figure 8: Caption and foil distribution of Change of State - Pre-State sub-phases before and after Amazon Mechanical Turk validation process.

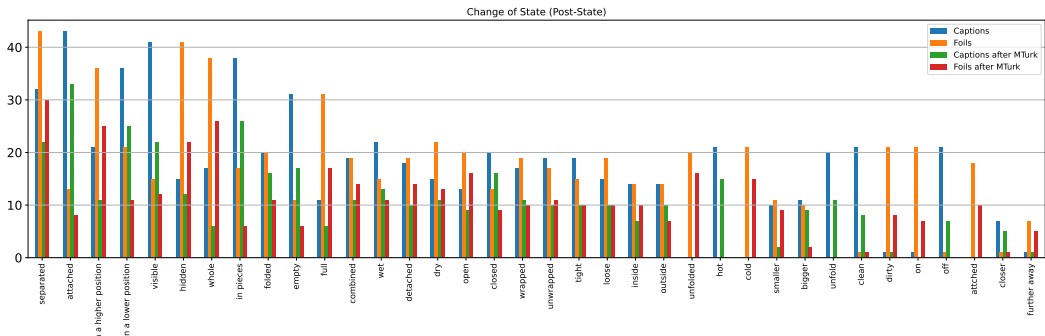

Figure 9: Caption and foil distribution of Change of State - Post-State sub-phases before and after Amazon Mechanical Turk validation process.

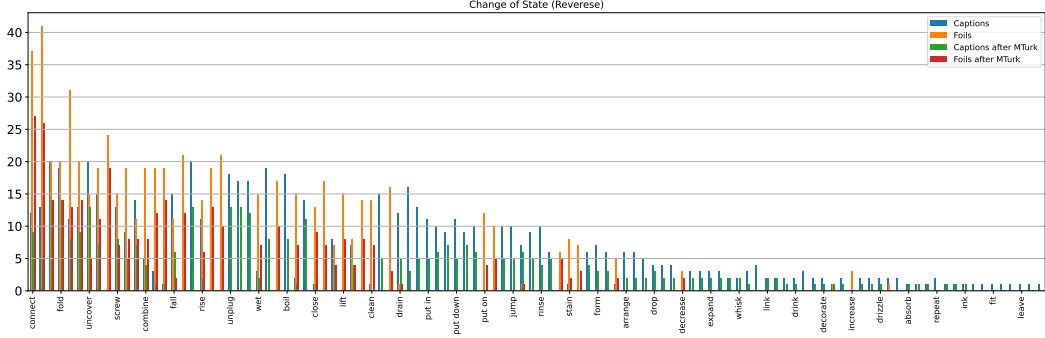

Figure 10: Caption and foil distribution of Change of State - Reverse test before and after Amazon Mechanical Turk validation process.

**Results** In Table 5 we show the statistics of the internal validation process for the proficiency tests. Also in this case, we check for potential distributional biases, measuring the Jensen-Shannon divergence of the word frequency distribution before and after the validation. We do not observe any significant change (see column *JS* and *JS Val.* in Table 5). The majority of the proficiency tests pass the internal manual validation, accounting for a total of 7182 (84.89%) of the original instances.

Finally, in Table 6, we summarise the statistics for VILMA combining the validation of proficiency and main tests. For our experiments, we only rely on samples where both the main test and corresponding proficiency test item have passed the validation.

Table 5: Manual internal validation results for proficiency tests in VILMA. *#Inst.*: number of instances for linguistic phenomenon. *#Valid (%)*: number (percent) of cases for which the annotator has chosen the caption; *#Lex.It.*: number of phrases or lexical items in the vocabulary that differ between foils and captions; *JS*: Jensen-Shannon divergence between foil-caption distributions for all instances in the whole subtest; *JS Val.*: Jensen-Shannon divergence between foil-caption distribution for the valid instances of the subset, after sub-sampling; $\alpha$: Krippendorff's $\alpha$ coefficient computed over all the instances; $\alpha$ *valid*: Krippendorff's $\alpha$ coefficient computed over the *Valid* instances.

| Test | Subtest | #Inst. | #Valid (%) | #Lex.it. | JS | JS Val. |
|---|---|---|---|---|---|---|
| **Change of State** | *All* | 624 | 412(66.03) | 293 | 0.391 | 0.372 |
| **Action Counting** | *Easy* | 959 | 939(97.91) | 9 | 0.234 | 0.235 |
| | *Difficult* | 895 | 884(98.77) | 11 | 0.224 | 0.226 |
| **Rare Actions** | *Action Replacement* | 978 | 940(96.11) | 0 | 0.335 | 0.334 |
| | *Object Replacement* | 972 | 907(93.31) | 0 | 0.342 | 0.342 |
| **Spatial Relations** | *Prepositions* | 708 | 633(89.41) | 59 | 0.239 | 0.241 |
| **Situation Awareness** | *Action Replacement* | 1000 | 837(83.70) | 127 | 0.108 | 0.108 |
| | *Actor Swapping* | 452 | 394(87.17) | 54 | 0.102 | 0.101 |
| **Overall** | | 8460 | 7182(84.89) | | | |

Table 6: VILMA statistics after Amazon Mechanical Turk and internal validation. *#Inst.*: number of instances for linguistic phenomenon. *#Valid Prof.*: number of valid cases in the proficiency test for which the annotator has chosen the caption; *#Valid Test.*: number of valid cases in the subtest for which 2 out of 3 annotators have chosen the caption; *#Both. Valid(%).*: number (percent) of valid cases for which both, the proficiency and test and the test case, are valid.

| Test | Subtest | #Inst. | #Valid. Prof. | #Valid. Test | #Both. Valid.(%) |
|---|---|---|---|---|---|
| **Change of State** | *Action* | 624 | 412 | 466 | 314(37.82) |
| | *Pre-State* | | | 286 | 194(31.08) |
| | *Post-State* | | | 383 | 254(40.70) |
| | *Reverse* | | | 342 | 236(37.82) |
| **Action Counting** | *Easy* | 959 | 939 | 774 | 757(78.94) |
| | *Difficult* | 895 | 884 | 682 | 675(75.42) |
| **Rare Actions** | *Action Replacement* | 978 | 940 | 781 | 751(76.79) |
| | *Object Replacement* | 972 | 907 | 739 | 692(71.19) |
| **Spatial Relations** | *Prepositions* | 708 | 633 | 436 | 393(55.51) |
| **Situation Awareness** | *Action Replacement* | 1000 | 837 | 838 | 704(70.40) |
| | *Actor Swapping* | 452 | 394 | 207 | 207(45.80) |
| **Overall** | | 8460 | 7182 | 5934 | 5177(61.19) |

## C  BENCHMARK CREATION

VILMA is intended as a zero-shot benchmark for Video-Language Models, divided into a number of main tests, each of which probes a model's capabilities in a specific phenomenon related to temporal reasoning and grounding. Main tests can be divided into sub-tests. Each main test is accompanied by a proficiency test, which probes the model's capabilities on a simpler task, which is considered a prerequisite to solving the main task.

### C.1  PROFICIENCY TESTS

We designed the proficiency tests to assess the ability of VidLMs' to solve simpler visio-linguistic tests that do not require strong temporal modelling. We consider proficiency in these tests to be an essential prerequisite for a VidLM to effectively tackle the main tests. A model succeeding on the main test but failing its corresponding proficiency test is a cause for concern: the model might rely on

signals within modalities that are easier to exploit instead of using the dynamic temporal information – likely due to the model's pretraining biases.

**Data sources.** Since proficiency tests supplement the main tests, we create them from the same data instances used to develop the samples for the corresponding main test.

**Foiling method.** We employ a consistent approach to create caption-foil pairs for all proficiency tests. When a proficiency test requires a model to identify objects or actions, or to verify the existence of an entity in the visual modality, we follow these steps: first, we use spaCy's[10] dependency parser to localise the target phrases. Target phrases can differ according to the main test's objective (e.g. nouns for Spatial Relations and verbs for Rare Actions). Then, we generate foil candidates by masking the relevant element (e.g. nouns) in the original sentence and predict the masked token using a Masked Language Modelling (MLM) by using either RoBERTa or T5 (t5-large). Then, we select the three most probable tokens from the model's output to create three candidate foil captions.

To ensure the quality of the foils, we employ a two-step procedure. In the first step, we use an *ALBERT*[11] model finetuned on Natural Language Inference (NLI). Given a caption and a foil, we expect a valid foil to not be true of the video. If the model predicts the foil to be entailed by the caption (E), we discard the sample. If the model predicts the foil to be neutral (N) or contradictory (C) with respect to the caption, we accept as a valid foil and proceed with the second step.

In the second step, we compute the GRUEN score via a BERT model finetuned on the Corpus of Linguistic Acceptability (CoLA). GRUEN (Zhu & Bhat, 2020) is a learned metric originally intended for use in Natural Language Generation, which returns a score based on an aggregate of Grammaticality, non-Redundancy, Discourse focus, Structure and coherence scores. If some sample has a GRUEN score lower than a certain threshold (e.g. 80%) we reject the sample, as it is not a valid foil.

For a candidate foil to be considered valid, it must pass both the NLI and GRUEN assessments together. If multiple candidates for a given sample pass both tests, we select the foil-caption pair that has the highest GRUEN score.

As a result, each instance in any main test has one caption-foil proficiency test pair. If none of the candidate sentences pass both steps, we discard that instance.

## C.2 ACTION COUNTING

The **Action Counting** test aims to probe the ability of models to accurately count the occurrences of actions within a given input video. Distinct from its image-based counterpart in the prior work VALSE (Parcalabescu et al., 2022), this test requires spatio-temporal reasoning, presenting a novel and interesting challenge.

**Data sources.** We use the QUVA dataset (Runia et al., 2018), comprising 100 videos. Within each video, every occurrence of the target action is annotated with a corresponding frame number, specifying the end of each action. The QUVA dataset lacks any textual annotations. Consequently, we curate multiple textual templates per video, incorporating a placeholder for the numerical value (<number>). Emulating the approach in VALSE, our templates incorporate the term *exactly* to indicate precise counting (e.g., someone performs exactly <number> push-ups). We take care to avoid overly specific terms, opting for more general descriptors (e.g., *lifting weights* instead of *skull-crushers arm exercise*). A native English speaker checked the manually collected templates and fixed potential syntax errors in them. We set the videos' frame per second rate to 30, since VideoCLIP (Xu et al., 2021) only works with 30-FPS videos.

**Foiling method.** To create captions and foils, we replace the number placeholder with the correct numerical value and an incorrect one. We discard all instances with counts exceeding a predetermined threshold $T_c$, set at 10. For the Action Counting test, we created the **easy** and the **difficult** subtests. In the **easy** subtest, we deliberately opt for small numbers $C \in \{1, 2, 3\}$ in the captions. The choice of

---

[10]spacy.io/

[11]https://huggingface.co/ynie/albert-xxlarge-v2-snli_mnli_fever_anli_R1_R2_R3-nli

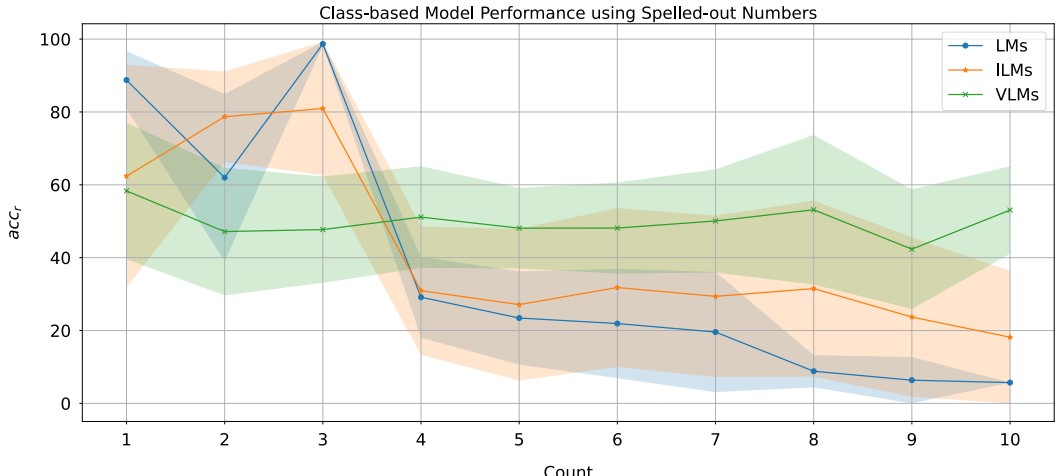

Figure 11: Categorical evaluation on the counting main tests. We simplify this analysis by computing average performances for each model category, the unimodal LMs, ILMs and VidLMs. The standard deviation values are illustrated with the colour filled areas.

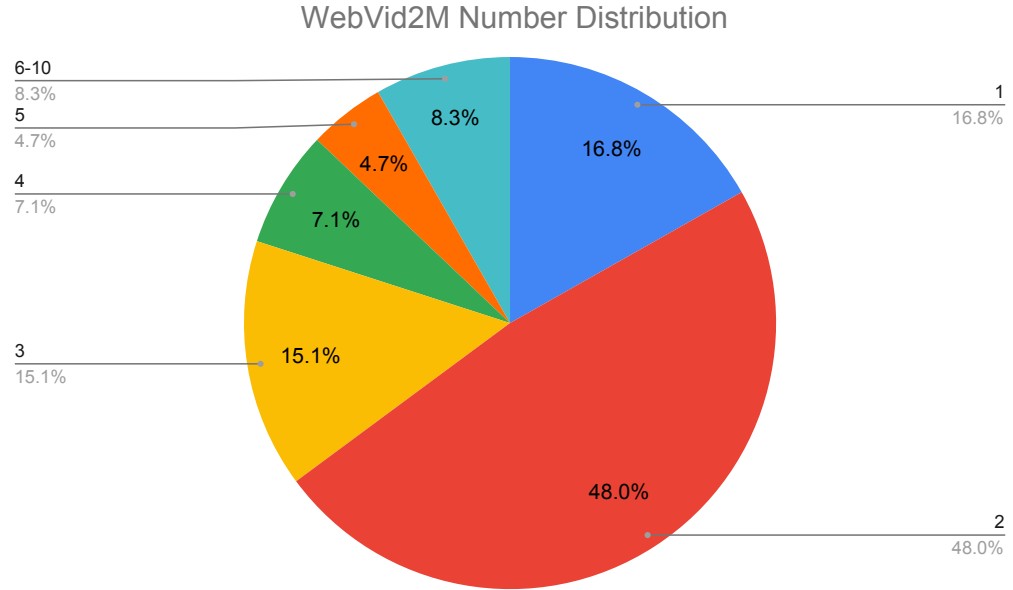

Figure 12: WebVid2M dataset (Bain et al., 2021) number distribution. The indefinite articles (a/an) are opted out. Numbers 6, 7, 8, 9 and 10 are merged into single category *6-10*.

these small numbers aligns with the notion that models frequently encounter such quantities during pretraining (see Figure 12), making them more recognisable and interpretable. In the **difficult** subtest, by contrast, we favour these same small numbers in the foils. This presents a challenging test for VidLMs as it tests the models' ability to overcome any bias towards numbers frequently encountered during pretraining. In this way, we aim to assess the models' true abilities to handle Action Counting tests in diverse contexts.

**Proficiency Test.** In the proficiency test, we assess how well the models recognise the actions repeated in the videos. To create the proficiency captions, we remove number-specific phrases. For instance, we change "a man performs exactly <number> push-ups." to "a man performs push-ups". To create proficiency foils, we implement a procedure that has 4 main stages:

1. We mask the verb phrases and generate text for the masked spans using T5 (t5-large) (Raffel et al., 2020). To obtain the initial foil candidates, we filter out generations that include personal pronouns (e.g. I, they, etc.) and conjunctions (e.g. and, but). We then perform GRUEN and NLI filtering (Zhu & Bhat, 2020). Similar to the other tests' proficiency test, we discard candidates that have a GRUEN score lower than a threshold of $0.80$ and the *entail* the proficiency caption. As a last step, we perform manual intervention and discard implausible foil candidates.

2. We mask the subject and noun phrases in the captions and then we repeat the first step using RoBERTa (Liu et al., 2019) for the examples that do not have a single foil. We repeat the first stage's GRUEN/NLI and manual filtering steps again.

3. For the videos without any valid foils, we randomly sample captions from the other videos. We restrict this sampling process categorically: For the exercise videos, we only sample captions from other exercise videos. We exclude the captions that comprise "*the same exercise*" phrase. We then replace the subject phrases with the ground-truth caption's subject phrase to make it similar to the true caption. We perform an NLI filtering as a final step to finalise the foil candidates.

4. To obtain the foil, we randomly sample from the candidate set.

We employ this 4-stage procedure because of the captions' degree of specificity for some examples. For instance, if we mask the verb or noun phrases of the sentence "*a man performs push-ups.*", LMs naturally fail to come up with different phrases. We can observe the same phenomena for sentences "*a kid jumps on a trampoline*" and "*somebody pushes a button*".

**In-Depth Results.** Table 7 shows the model results on the Action Counting tests.

UNIMODAL RESULTS. We notice a notable bias among the unimodal baselines, specifically LMs, towards smaller numbers. This inclination aligns with our expectations, considering that these models lack visual input processing capabilities. Predictably, their performance in the combined setting closely mirrors that of a random baseline.

IMAGE-LANGUAGE MODEL RESULTS. Unlike LMs, ILMs achieve a good performance in the proficiency tests, demonstrating their proficiency in visual comprehension. That being said, they are incompetent to count actions because of their nature. Interestingly, BLIP2 heavily favours small numbers like ILMs, indicating that it overlooks the visual modality to a significant degree.

VIDEO-LANGUAGE MODEL RESULTS. We tested several kinds of VidLMs and CLIP4Clip (Luo et al., 2022) achieved the best results in the proficiency subtest (P), significantly outperforming the other models. On the other hand, when evaluating performance in the main test (T), Merlot Reserve (Zellers et al., 2022) took the lead, giving the best results. However,CLIP4Clip gives the best results in the combined results (P+T) among all models we tested for the Action Counting test. Figure 11 illustrates their overall count-specific performance. As it can be seen from this figure and Table 7, their average performance is close to a random baseline, revealing that these models are far away from being proficient to excel such a challenging spatio-temporal grounding problem.

**Test Examples.** In Figure 13 we show some sample validated examples from the **action counting** main tests.

Table 7: Action Counting subtest results using pairwise accuracy ($acc_r$) metric. P, T and P+T stand for the scores achieved on the proficiency tests, the main tests only and the combined tests, respectively.

| Model | Easy | | | Difficult | | | All | | |
|---|---|---|---|---|---|---|---|---|---|
| | P | T | P+T | P | T | P+T | P | T | P+T |
| Random | 50.00 | 50.00 | 25.00 | 50.00 | 50.00 | 25.00 | 50.00 | 50.00 | 25.00 |
| GPT-2 | 50.46 | 70.67 | 35.54 | 50.07 | 33.78 | 18.67 | 50.30 | 53.30 | 27.60 |
| OPT | 55.48 | **93.79** | 52.44 | 56.89 | 10.67 | 6.96 | 56.20 | 54.60 | 31.00 |
| CLIP | 91.28 | 51.65 | 45.71 | 89.63 | 50.07 | 46.67 | 90.50 | 50.90 | 46.20 |
| BLIP2 | 80.71 | **93.79** | **75.03** | 81.04 | 10.37 | 8.44 | 80.90 | 54.50 | 43.70 |
| ClipBERT | 56.80 | 12.42 | 7.27 | 56.00 | **91.26** | 51.26 | 56.40 | 49.60 | 28.00 |
| UniVL | 71.60 | 47.29 | 31.70 | 71.70 | 46.81 | 40.00 | 73.40 | 43.60 | 32.20 |
| VideoCLIP | 78.60 | 31.57 | 25.36 | 79.70 | 62.96 | 49.04 | 79.10 | 46.40 | 36.50 |
| FiT | 84.81 | 52.44 | 44.91 | 82.81 | 52.30 | 44.15 | 83.90 | 52.40 | 44.60 |
| CLIP4Clip | **91.55** | 76.35 | 69.62 | **90.81** | 25.33 | 23.70 | **91.20** | 52.30 | 47.97 |
| VIOLET | 73.45 | 50.86 | 40.42 | 75.41 | 50.37 | 37.33 | 79.60 | 50.60 | 36.50 |
| X-CLIP | 84.68 | 68.16 | 57.07 | 83.41 | 40.44 | 34.52 | 84.10 | 55.10 | 46.40 |
| MCQ | 81.37 | 30.65 | 28.01 | 81.33 | 72.44 | **56.59** | 81.40 | 50.40 | 41.50 |
| Singularity | 79.92 | 61.16 | 48.35 | 79.26 | 39.70 | 33.78 | 79.60 | 51.10 | 41.50 |
| UniPerceiver | 50.99 | 22.06 | 11.36 | 50.07 | 73.63 | 36.00 | 50.56 | 46.37 | 22.97 |
| Merlot Reserve | 83.62 | 53.37 | 44.39 | 84.74 | 58.96 | 49.63 | 84.15 | 56.01 | 46.86 |
| VindLU | 85.73 | 65.13 | 57.60 | 83.11 | 35.56 | 27.70 | 84.50 | 51.20 | 43.50 |
| InternVideo | 90.62 | 63.94 | 48.74 | 89.63 | 43.56 | 48.74 | 90.15 | 54.33 | **48.74** |
| mPLUG-2 | 57.10 | 53.70 | 31.70 | 58.40 | 45.30 | 23.10 | 57.70 | 49.70 | 27.70 |
| Otter | 59.45 | 90.75 | 54.16 | 59.41 | 14.67 | 7.26 | 59.43 | 52.71 | 30.71 |
| Video-LLaMA | 85.47 | 66.84 | 56.01 | 83.56 | 44.44 | 37.48 | 84.57 | **56.28** | 47.28 |

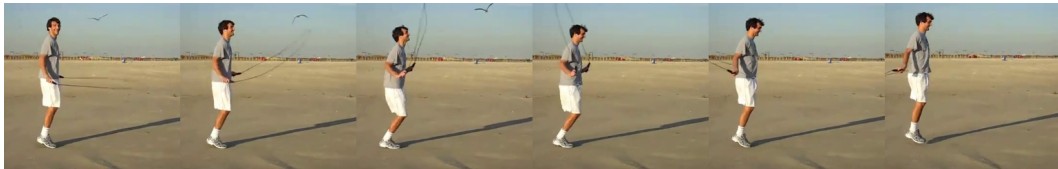

**Proficiency Test:** a man skips / climbs a rope.
**Main Test:** a man skips rope exactly three / nine times.

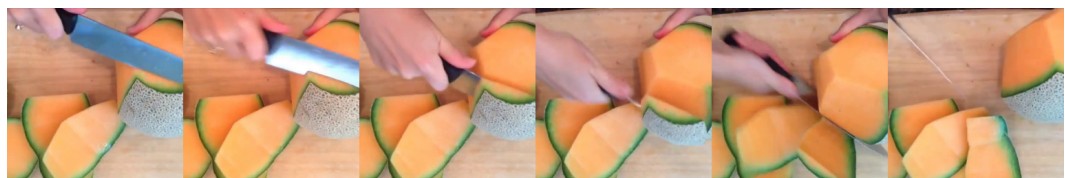

**Proficiency Test:** someone peels a melon / lemon.
**Main Test:** someone peels a melon in exactly two / five moves.

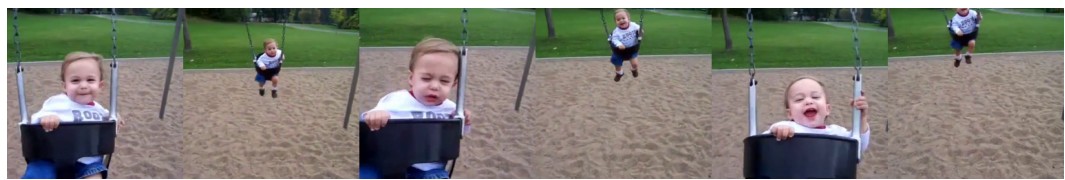

**Proficiency Test:** a toddler in a playground swings on a swing / rope.
**Main Test:** a toddler in a playground swings on a swing exactly three / ten times.

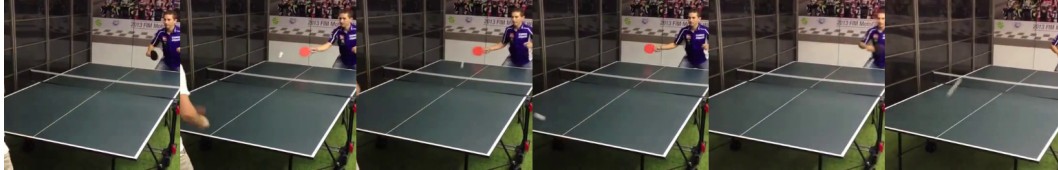

**Proficiency Test:** each table tennis player hits / catches the ball.
**Main Test:** each player hits the ball exactly three / five times using their rackets.

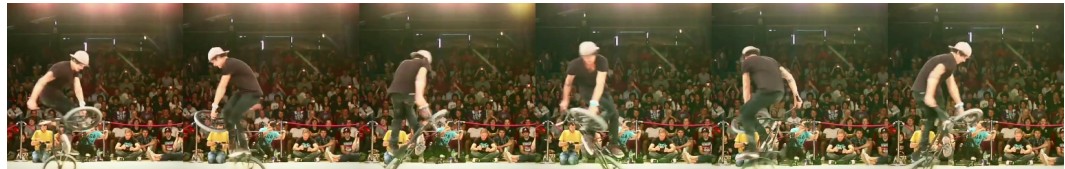

**Proficiency Test:** a performer whirls / walks around.
**Main Test:** a man on a bike spins exactly two / five times.

Figure 13: Sample instances from the **action counting** tests. We only show examples from the easy subtests, since larger counts become difficult to perceive, when videos are represented in limited number of frames.

## C.3 SITUATION AWARENESS

In the **Situation Awareness** test, we devise two subtests known as **Actor Swapping** and **Action Replacement**. With the Action Replacement subtest, we assess the effectiveness of video-language models in distinguishing diverse actions within a particular video stream. This evaluative approach involves juxtaposing two sentences that differ only in the action verbs used. The model's ability to recognise and distinguish different activities is assessed by comparing the model's consequent scores ascribed to these statements.

The Actor Swapping subtest is intended to assess a model's ability to recognise key individuals in the footage. The evaluation involves presenting a pair of sentences in which the actors and actants are switched, resulting in a different syntactic configuration. The model's ability to recognise and identify the actors engaged is demonstrated by a comparison of the scores assigned to these statements.

These evaluative tools are critical for assessing model performance because they provide for a more sophisticated understanding of the model's capacity to distinguish semantic roles and relational dynamics within video-language situations.

**Data sources.** For the SA subtests, we leverage the VidSitu (Sadhu et al., 2021) dataset, a vast collection of 10-second videos from movies that show intricate scenarios annotated with verbs, semantic roles, entity co-references, and event relations at 2-second intervals.

To create captions for Situation Awareness subtests, we use ChatGPT (OpenAI, 2021). We give the prompt below to ChatGPT with raw sentences that we obtained from the VidSitu (Sadhu et al., 2021) dataset:

*I want you to act as an English spelling corrector and improver. I will speak to you in English and you will answer in the corrected and improved version of my text, in English. I want you to replace my simplified A0-level words and sentences with more beautiful and elegant, upper-level English words and sentences. Keep the meaning same, but make them more literary. I want you to only reply with the corrections, the improvements, and nothing else, do not write explanations. Your responses should be enumerated. Each sentence is separated by a dot (.). My sentences are: <sentences>*

We evaluated the readability scores of generated sentences with the Flesch-Kincaid (Kincaid et al., 1975) and Flesch Reading Ease (Flesch, 1948) methods. According to the Flesch-Kincaid and Flesch Reading Ease methods, the produced text received scores of 4.54 and 83.27, representing grade levels of 5 and 6, respectively.

**Foiling method.** We construct alternative foils for each caption by picking the top 32 most probable tokens from RoBERTa *RoBERTa-base*[12] outputs. We conduct a dual-phase review to determine their validity. First, using an *ALBERT*[13] model, we use Natural Language Inference (NLI) to evaluate the foils' conformity to video content. We reject those identified as entailment and keep those labelled as neutral or contradiction. Following that, we assess grammatical integrity with the GRUEN score and eliminate foils with less than an 80% GRUEN score. Foils must succeed in both NLI and GRUEN trials, ensuring contextual in-congruence and linguistic coherence.

**Proficiency Test.** In the Proficiency test of **Situation Awareness**, our primary focus is on **object identification**, which plays a critical role in assessing the model's ability to recognise actions and actors. This emphasis on object identification is essential because it serves as the cornerstone for understanding actions within given scenarios and identifying the individuals or entities involved, which are essential aspects of situational comprehension. Our approach involves masking objects based on the transitivity of verbs: when a verb takes an object, we substitute it with a counterfactual created by RoBERTa (Liu et al., 2019), allowing us to assess the model's understanding of the object's role in actions. Conversely, if a verb cannot have a direct object, we mask the subject, ensuring a comprehensive assessment of the model's capability to identify actors. Object identification, in this context, enables a holistic understanding of scenes, helping the model comprehend the broader context and relationships between elements in dynamic scenarios, aligning perfectly with the objectives of the Situation Awareness test.

---

[12]https://huggingface.co/roberta-base
[13]https://huggingface.co/ynie/albert-xxlarge-v2-snli_mnli_fever_anli_R1_R2_R3-nli

Table 8: Situation Awareness subtests results using pairwise accuracy ($acc_r$) metric. P, T and P+T stand for the scores achieved on the proficiency tests, the main tests only and the combined tests, respectively.

| Model | Action Replacement | | | Actor Swapping | | | All | | |
|---|---|---|---|---|---|---|---|---|---|
| | P | T | P+T | P | T | P+T | P | T | P+T |
| Random | 50.00 | 34.42 | 17.21 | 50.00 | 50.00 | 25.00 | 50.00 | 37.96 | 18.98 |
| GPT-2 | 43.03 | 67.47 | 31.68 | 49.76 | 63.77 | 31.88 | 44.57 | 66.63 | 31.72 |
| OPT | 50.14 | _71.88_ | 38.49 | 57.00 | _69.57_ | 39.61 | 51.70 | _71.35_ | 38.75 |
| CLIP | 70.74 | 45.03 | 33.66 | 71.98 | 47.34 | 33.82 | 71.02 | 45.55 | 33.70 |
| BLIP2 | 72.30 | **78.12** | **57.24** | _77.29_ | 66.18 | _50.72_ | 73.44 | **75.41** | **55.76** |
| ClipBERT | 53.41 | 55.11 | 29.69 | 56.52 | 63.29 | 39.61 | 54.12 | 56.97 | 31.94 |
| UniVL | 53.98 | 44.46 | 23.86 | 49.28 | 54.11 | 25.12 | 52.91 | 46.65 | 24.15 |
| VideoCLIP | 62.78 | 37.36 | 22.59 | 57.97 | 50.72 | 32.85 | 61.69 | 40.40 | 24.92 |
| FiT | 68.47 | 38.64 | 27.56 | 74.40 | 44.93 | 34.78 | 69.81 | 40.07 | 29.20 |
| CLIP4Clip | _73.15_ | 46.59 | 35.51 | 76.33 | 57.49 | 44.93 | _73.87_ | 49.07 | 37.65 |
| VIOLET | 69.32 | 41.19 | 29.69 | 73.43 | 55.56 | 42.03 | 70.25 | 44.46 | 32.49 |
| X-CLIP | 64.91 | 43.32 | 30.68 | 58.94 | 50.24 | 32.37 | 63.56 | 44.90 | 31.06 |
| MCQ | 65.20 | 33.10 | 22.44 | 73.43 | 50.72 | 39.61 | 67.07 | 37.10 | 26.34 |
| Singularity | 67.05 | 38.78 | 27.70 | 74.88 | 48.31 | 38.65 | 68.83 | 40.94 | 30.19 |
| UniPerceiver | 52.13 | 29.12 | 14.35 | 49.28 | **86.47** | 44.44 | 51.48 | 42.15 | 21.19 |
| Merlot Reserve | 68.89 | 30.97 | 21.16 | 76.33 | 51.69 | 39.61 | 70.58 | 35.68 | 25.36 |
| VindLU | 69.46 | 39.63 | 29.40 | 74.40 | 48.31 | 37.68 | 70.58 | 41.60 | 31.28 |
| InternVideo | 70.88 | 39.20 | 28.12 | 73.91 | 47.34 | 34.30 | 71.57 | 41.05 | 29.53 |
| mPLUG-2 | 47.60 | 34.70 | 18.32 | 56.50 | 46.40 | 32.40 | 49.60 | 37.40 | 21.50 |
| Otter | 58.10 | 39.63 | 23.86 | 59.42 | 62.32 | 34.78 | 58.76 | 50.98 | 29.32 |
| Video-LLaMA | **77.56** | 67.61 | _53.55_ | **80.19** | 64.73 | **55.56** | **78.15** | 66.96 | _54.01_ |

**In-Depth Results.** We share detailed results of **Situation Awareness** subtests in Table 8.

UNIMODAL RESULTS. In both the Action Replacement and Actor Swapping subtests, we observe that GPT-2 and OPT tend to perform slightly below or near the random baseline in the proficiency test. This intriguing pattern hints at the inherent challenge these models face when tasked with grasping fundamental knowledge about objects or actions within the scenes. However, when it comes to the Main Tests, a noteworthy distinction emerges: GPT-2 significantly surpasses the random baseline, demonstrating its ability to comprehend and execute the task to a certain degree. In contrast, OPT performs much worse in the Main Test, possibly due to challenges in capturing crucial contextual details and disparities in data distribution.

IMAGE-LANGUAGE MODEL RESULTS. In every subtest, CLIP and BLIP2 perform much better than random performance, demonstrating their exceptional capacity to understand and work with both textual and visual information. Their design, which integrates cutting-edge vision and language models, enables them to perform well in tasks that call for comprehension and content creation in both modalities.

VIDEO-LANGUAGE MODEL RESULTS. Most Video-Language Models perform slightly better than random in these subtests, indicating some level of comprehension of video content, likely due to their ability to process temporal information and recognise visual cues. However, they are generally outperformed by Image-Language Models which may be attributed to the latter's specialised focus

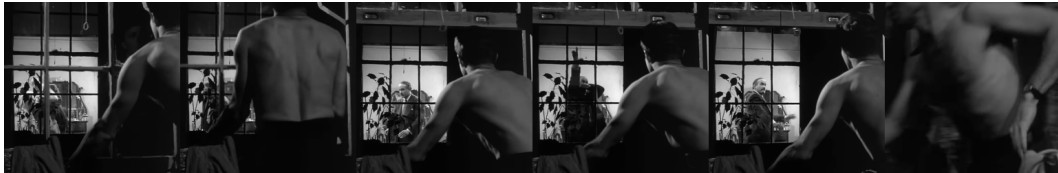

**Proficiency Test:** A shirtless man opens the window / door hurriedly.
**Main Test:** A shirtless man opens / smashes the window hurriedly.

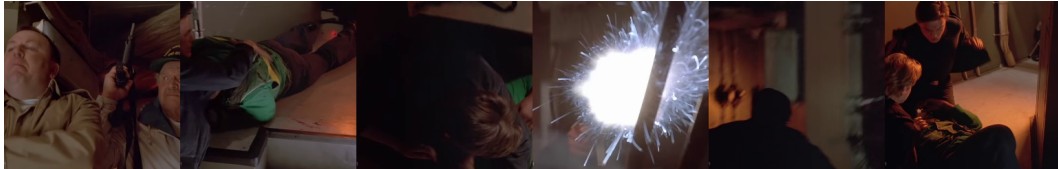

**Proficiency Test:** The man in a navy blue coat drags the man in the green coat off the ledge / ground.
**Main Test:** The man in a navy blue coat drags / tosses the man in the green coat off the ledge.

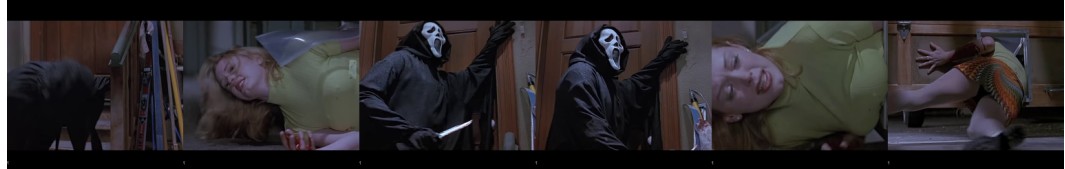

**Proficiency Test:** A girl, wearing a yellow top, forcefully pushes her body / away.
**Main Test:** A girl, wearing a yellow top, forcefully pushes / covers her body.

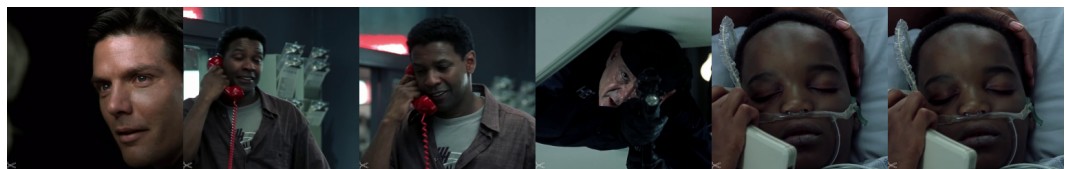

**Proficiency Test:** A man, dressed in a black outfit, aims his gun at a man in a brown shirt / SUV.
**Main Test:** A man, dressed in a black outfit, aims / discharges his gun at a man in a brown shirt.

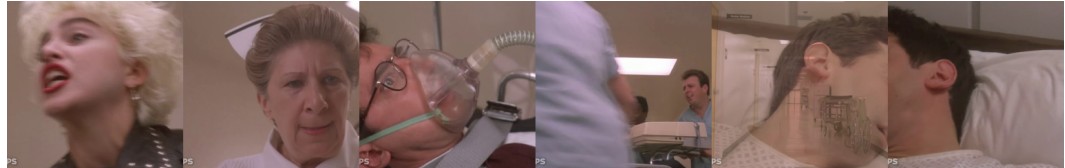

**Proficiency Test:** A man with a face mask breathes oxygen with difficulty / goggles.
**Main Test:** A man with a face mask breathes / measures oxygen with difficulty.

Figure 14: Sample instances from the **Situation Awareness (Action Replacement)** test.

on multimodal understanding, enabling them to excel in tests that require both textual and visual reasoning. Video-Language Models might need further refinement to fully leverage the richness of video data and match the performance of their image-based counterparts.

**Test Examples.** In Figure 14 and Figure 15, we show some sample validated examples from the **Situation Awareness – Action Replacement and Actor Swapping** subtests.

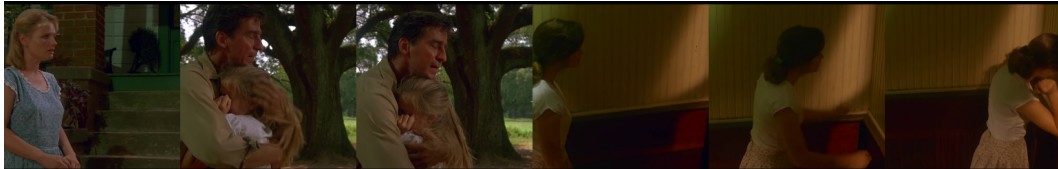

**Proficiency Test:** The man in a tan coat clasps the woman with blonde hair / locks outside.
**Main Test:** The man in a tan coat / woman with blonde hair clasps the woman with blonde hair / man in a tan coat outside.

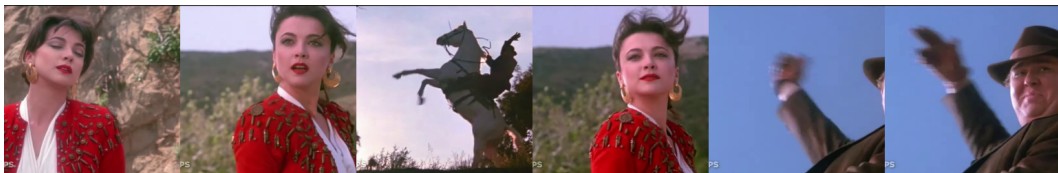

**Proficiency Test:** The woman in red looks upward at the man in a hat / wheelchair.
**Main Test:** The woman in red / man in a hat looks upward at the man in a hat / woman in red.

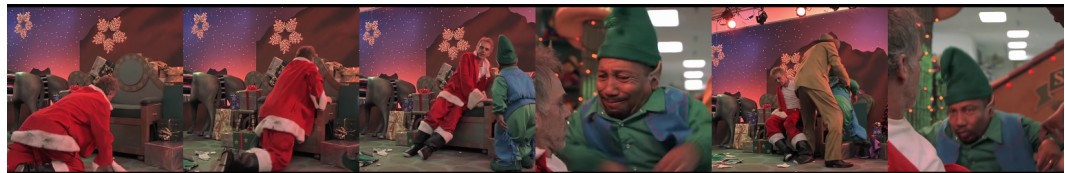

**Proficiency Test:** A gentleman dressed in a tan suit pulls a man in a green shirt by the arms / collar.
**Main Test:** A gentleman dressed in a tan suit / man in a green shirt pulls a man in a green shirt / gentleman dressed in a tan suit by the arms.

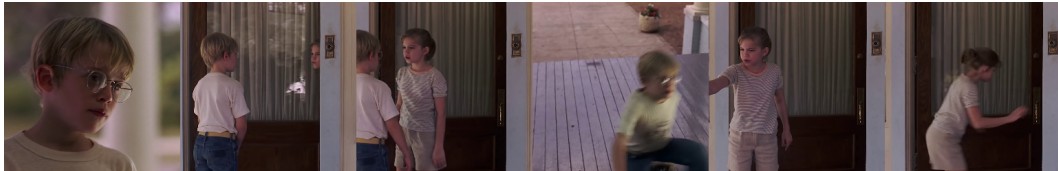

**Proficiency Test:** The girl with the ponytail suddenly pushes the boy with glasses / her.
**Main Test:** The girl with the ponytail / The boy with glasses suddenly pushes the boy with glasses / the girl with the ponytail.

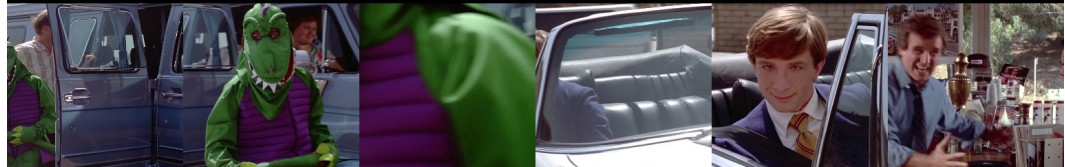

**Proficiency Test:** A young man sitting in a car notices a person wearing a dinosaur costume near a van / tree and a parked car.
**Main Test:** A young man sitting in a car / person wearing a dinosaur costume notices a person wearing a dinosaur costume / young man sitting in a car near a van and a parked car.

Figure 15: Sample instances from the **Situation Awareness (Actor Swapping)** test.

## C.4 CHANGE OF STATE

The **Change of State** test explores VidLMs' capacity to recognise and distinguish various sub-phases within actions, especially those involving Change of State (CoS) verbs. We also assess their ability to align these sub-phases across both textual and visual modalities. Aligning CoS actions, pre-state and post-states across modalities presents a unique challenge. While these states are typically implicit in text, they are explicit in the visual modality. The sequence of sub-phase events represents an instance of common-sense knowledge that is often not explicitly encoded in text, with pre- and post-states frequently assumed as logical outcomes of an action, or necessary preconditions for that same action to happen. Unimodal models may struggle to meaningfully model these sequences. In contrast, multimodal models equipped with visual and temporal perception modules such as VidLMs can bridge this knowledge gap, effectively transferring information encoded in the visual domain to the language domain and vice versa by successfully modelling a cross-modal space.

**Data sources.** In total, we collect 624 caption-video pairs from five different datasets:

1. Something-Something V2 (Goyal et al., 2017a) is a collection of 220,847 labelled video clips of humans performing pre-defined, basic actions with everyday objects. The dataset contains 168,913 videos in the training set, 24.777 in the validation set, and 27,157 in the test set. For our purposes, we focus solely on the validation set, from which we extract 312 caption-video pairs.

2. YouCook2 (Zhou et al., 2018) is an instructional video dataset with 2000 untrimmed videos from 89 cooking recipes. Each video has temporal annotations and imperative English descriptions. Videos are from YouTube, shot in third-person view, and depict unconstrained cooking scenarios worldwide. From this dataset we extract 154 caption-video pairs.

3. COIN (Tang et al., 2019) is an instructional dataset with 11,827 videos covering 180 tasks across 12 domains. It utilises a hierarchical structure with three levels: domain, task, and step. Domains include nursing, vehicles, gadgets, and more. Tasks are linked to domains (e.g., "replace a bulb" is linked to "electrical appliances"). Steps further detail tasks, like "remove the lampshade" for "replace a bulb.". From this dataset we extract 281 caption-video pairs.

4. RareAct (Miech et al., 2020) is a video dataset of unusual actions. It contains 122 different actions obtained by combining verbs and nouns rarely co-occurring together in the large-scale textual corpus from HowTo100M (Miech et al., 2019), but that frequently appear separately. From this dataset we extract 21 caption-video pairs.

5. STAR (Wu et al., 2021) is a benchmark for Situated Reasoning that provides challenging question-answering tasks, symbolic situation descriptions, and logic-grounded diagnosis via real-world video situations. It aims to capture the present knowledge from surrounding situations and reason accordingly. The dataset consists of four question types for situated reasoning: Interaction, Sequence, Prediction, and Feasibility. In order to construct captions from this dataset we extract the subject and the object from the question and combine them with the verb contained in the correct answer. From this dataset we extract 44 caption-video pairs.

**Foiling method.** For each subtest in the Change of State test, we generate distinct caption-foil pairs. However, the initial process is shared across all settings. We select candidates CoS verbs from a codebase initially developed by Warstadt et al. (2019) and later expanded by Warstadt et al. (2020). The codebase[14] includes over 3000 lexical items annotated by human experts with grammatical features. From this set of items, we select 49 verbs labelled as *change of state*. Alongside their grammatical features, we also retain the *initial state*, whenever available. Finally, for each CoS verb we retrieve a set of candidate *final states* (or post-states) as well as of antonyms. Specifically, we leverage ConceptNet (Speer et al., 2017), WordNet (Miller, 1995), and NLTK (Bird, 2006) to collect a set of appropriate candidates. Afterwards, we select the most appropriate one through manual validation. Consequently, each *change of state* can be represented as a 4-tuple consisting of the *pre-state*, the *CoS verb*, the *post-state*, and the *reverse CoS verb* (e.g., <*"open", "to close", "closed", "to open"*>). We use the *CoS verbs* as targets when parsing the textual data of various existing VidL

---

[14]https://github.com/alexwarstadt/data_generation

datasets. If any of the selected *CoS verbs* appears as a verb in a sentence, we retrieve that sentence along with its corresponding video item and metadata, if available. Each of caption-foil pairs within the subtests of the Change of State test is generated based on a template that specifies the linear order of the constituents in the sentence. The script for the pairs generation is available in the VILMA GitHub repository. For the generated caption, we define the following templates:

1. Action caption template: *"Someone <change-of-state-verb> the <object>."* for transitive change-of-state verbs, *"The <subject> <change-state-verb>."* for intransitive ones;

2. Pre-State caption template: *"Initially, the <subject/object> is <pre-state>.";*

3. Post-State caption template: *"At the end, the <subject/object> is <post-state>.";*

4. Reverse caption template: *"Initially, the <object> is <pre-state>. Then someone <change-state-verb> the <object>. At the end, the <object> is <post-state>."* for transitive change-of-state-verbs. *"Initially, the <subject> is <pre-state>. Then the <subject> <change-state-verb>. At the end, the <subject> is <post-state>."* for intransitive ones.

To generate the corresponding foiled versions, we replace the target sub-phase with its "opposite" sub-phase retrieved from the extended codebase. Specifically, for the *action sub-phase*, we replace it with the "reverse" element; for the *pre-state sub-phase*, we replace the "pre-state" with the "post-state", and vice-versa for the *post-state sub-phase*; finally for the reverse-foiling, targeting *all of the sub-phases*, we swap "pre-state" and "post-state" as well as replace the "action" with the "reverse" element.

**Proficiency Test.** In the proficiency test for **Change of State**, our focus is on the **object identification** task. For each unique video, we mask either the subject or the object of the caption, depending on the transitivity of the verb. If the verb can take an object as an argument, we mask it and replace it with a counterfactual generated by RoBERTa (Liu et al., 2019). If the verb cannot take a direct object, we mask the subject of the sentence instead.

**In-Depth Results.** In Table 9 , we report the in depth-results obtained by the models on all of the Change of State subtests.

UNIMODAL RESULTS. In the Action and Reverse subtests (T), unimodal models perform close to the random baseline. In these settings, text-only LMs are unable to identify the foiled version by processing the textual modality alone. However, in the Pre-State and Post-State settings (T), the performance levels deviate from the random guess, with a preference for the foiled version in the former, and the caption in the latter. This observation supports our assumption: change-of-state verbs often occur together with their respective Post-State, while their preconditions (i.e., the Pre-State) are typically implied rather than explicitly stated in text corpora. This biased distribution is reflected in the performance of unimodal LMs in these two subtests, where the model tends to blindly attribute a lower perplexity score to the sentence containing the post-state condition. However, if we take into account the combined results (P+T), all of the results shows that text-only LMs are generally unable to correctly solve the tests, despite GPT-2 achieving the best score on the board (T).

IMAGE-LANGUAGE MODEL RESULTS. The same general trend applies to ILMs, as well. Both ILMs achieve slightly higher results on the Action subtests, while results on the Reverse one remain mixed (T). The ability to integrate static visual information is not sufficient to correctly distinguish between an action unfolding in one temporal direction instead of the other. ILMs still exhibit a bias towards sentences containing the Post-State phase rather than those containing the Pre-State one. Results obtained on the combined (P+T) tests reveal the necessity of multimodal information in order to meaningfully solve the tests when compare to the Unimodal (P+T) results.

VIDEO-LANGUAGE MODEL RESULTS. Across the board, VidLMs exhibit some minor improvements over other types of models in the Reverse (T) subtest. When the unfolding of the action is completely explicit, VidLMs outperform unimodal and ILMs ones (i.e., best and second-best scores are obtained by InternVideo and Merlot Reserve, respectively). However, VidLMs second best result (Merlot Reserve) on the combined subtest (P+T) is matched by CLIP.

Table 9: Change of State results using pairwise accuracy ($acc_r$) metric. P, T and P+T stand for the scores achieved on the proficiency tests, the main tests only and the combined tests, respectively.

| Model | Action | | | Pre-State | | | Post-State | | | Reverse | | | All | | |
|---|---|---|---|---|---|---|---|---|---|---|---|---|---|---|---|
| | P | T | P+T | P | T | P+T | P | T | P+T | P | T | P+T | P | T | P+T |
| Random | 50.0 | 50.0 | 25.0 | 50.0 | 50.0 | 25.0 | 50.0 | 50.0 | 25.0 | 50.0 | 50.0 | 25.0 | 50.0 | 50.0 | 25.0 |
| GPT-2 | 18.5 | 51.3 | 11.8 | 19.1 | 40.2 | 8.8 | 16.1 | **65.8** | 11.0 | 18.2 | 52.5 | 11.4 | 18.0 | 52.4 | 10.8 |
| OPT | 24.2 | 54.1 | 17.5 | 23.2 | 37.6 | 7.7 | 22.4 | 59.8 | 15.0 | 22.5 | 40.2 | 11.4 | 23.1 | 48.0 | 12.9 |
| CLIP | 93.0 | 60.2 | 57.3 | 93.3 | 46.9 | 43.8 | 93.3 | 57.1 | 53.5 | 92.4 | 56.8 | 54.2 | 93.0 | 55.2 | 52.2 |
| BLIP2 | 75.2 | 56.4 | 41.4 | 73.2 | 40.2 | 29.9 | 74.8 | 65.0 | 47.6 | 75.0 | 47.0 | 33.5 | 74.5 | 52.1 | 38.1 |
| ClipBERT | 63.1 | 47.5 | 33.1 | 62.4 | 41.8 | 26.3 | 63.4 | 53.1 | 33.5 | 66.1 | 57.6 | 41.1 | 63.7 | 50.0 | 33.5 |
| UniVL | 81.5 | 58.0 | 46.5 | 80.4 | 47.4 | 34.5 | 81.5 | 55.9 | 47.2 | 81.8 | 55.9 | 43.6 | 81.3 | 54.3 | 43.0 |
| VideoCLIP | 48.7 | 51.0 | 25.8 | 50.5 | **52.6** | 32.5 | 46.1 | 51.2 | 19.7 | 53.8 | 48.3 | 25.9 | 49.8 | 50.8 | 25.9 |
| FIT | 93.0 | 56.7 | 52.2 | 93.3 | 42.8 | 38.1 | 94.1 | 53.9 | 52.0 | 91.5 | 55.1 | 48.7 | 93.0 | 52.1 | 47.8 |
| CLIP4Clip | 94.3 | 56.7 | 55.1 | **95.4** | 46.9 | 44.9 | 94.9 | 57.5 | 55.5 | 94.5 | 55.5 | 53.0 | 94.8 | 54.1 | 52.1 |
| VIOLET | 87.9 | 55.1 | 50.0 | 90.7 | 47.4 | 43.3 | 86.2 | 57.5 | 49.2 | 88.1 | 58.5 | 53.8 | 88.2 | 54.6 | 49.1 |
| X-CLIP | 85.7 | 51.3 | 45.9 | 85.0 | 49.0 | 39.2 | 87.0 | 55.1 | 50.8 | 85.2 | 55.5 | 48.3 | 85.7 | 52.7 | 46.0 |
| MCQ | 91.7 | 53.2 | 50.0 | 88.1 | 42.3 | 34.0 | 91.3 | 54.3 | 50.0 | 89.8 | 51.3 | 47.0 | 90.3 | 50.3 | 45.3 |
| Singularity | 93.6 | 61.1 | 56.7 | 91.8 | 44.3 | 40.7 | 94.5 | 60.6 | **57.5** | 91.5 | 52.1 | 46.2 | 92.8 | 54.6 | 50.3 |
| UniPerceiver | 68.5 | 43.3 | 27.1 | 63.9 | 43.8 | 23.2 | 69.7 | 53.1 | 41.3 | 67.8 | 44.1 | 25.0 | 67.5 | 46.1 | 29.1 |
| Merlot Reserve | 93.0 | 62.4 | **58.9** | 94.8 | 34.5 | 32.5 | 93.7 | 58.7 | 55.9 | 92.0 | 58.9 | 54.2 | 93.4 | 53.6 | 50.4 |
| VindLU | 85.7 | 63.1 | 54.1 | 84.0 | 46.9 | 39.7 | 84.7 | 51.2 | 45.3 | 87.3 | 49.1 | 43.2 | 85.4 | 52.6 | 45.6 |
| InternVideo | **95.5** | 60.5 | 55.1 | **95.4** | 47.4 | **55.1** | **95.7** | 55.9 | 55.1 | **95.8** | 64.4 | **55.1** | **95.6** | 57.7 | **55.1** |
| mPLUG-2 | 38.9 | 51.3 | 23.0 | 39.7 | 43.3 | 17.0 | 40.2 | 50.8 | 22.4 | 39.4 | 43.2 | 19.1 | 39.5 | 47.7 | 20.8 |
| Otter | 66.6 | 57.0 | 39.2 | 61.3 | 44.3 | 25.8 | 67.3 | 60.2 | 40.2 | 67.4 | 50.4 | 31.8 | 65.7 | 53.0 | 34.3 |
| Video-LLaMA | 82.2 | **65.0** | 51.9 | 80.9 | 45.9 | 39.7 | 81.9 | 62.2 | 48.4 | 80.1 | 58.5 | 44.1 | 81.4 | **59.0** | 46.8 |

VidLMs still struggle on the Pre-State subtest stressing their inability to align the visual and textual modalities. Even models pretrained with NLG tasks, such as UniVL, do not perform better than the random baseline. However, most of the VidLMs tend to closer approximate the random guessing (T) allowing us to hypothesise that the textual bias toward pre-state sentences exhibited by Unimodal and ILMs is slightly reduced.

On the other hand, this leads to lower results in the Post-State (T) subtest. Surprisingly, in the Action subtest (T), where VidLMs are not provided with the explicit textual Pre-State and Post-State information, models do not perform significantly better than ILMs.

However, in all of the subtests, when we consider the combined tests (P+T) we can better appreciate the information provided by the visual modality. Nevertheless, the performance gap between VidLMs and ILMs is neither pronounced nor consistent, with CLIP scoring the second-best result on the aggregated results for the combined metric (P+T), and Video-LLaMA and InternVideo showing only marginal improvement over the best ILM.

**Test Examples.** In Figure 16 - 19, we present some sample validated examples from the **Change of State – Action, Pre-State, Post-State and Reverse** subtests.

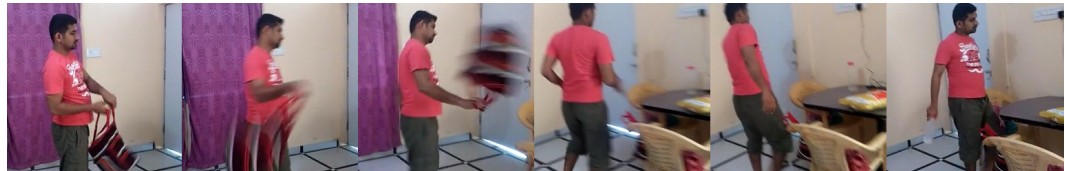

**Proficiency Test:** Someone throws the bag / ball.
**Main Test:** Someone throws / pulls the bag.

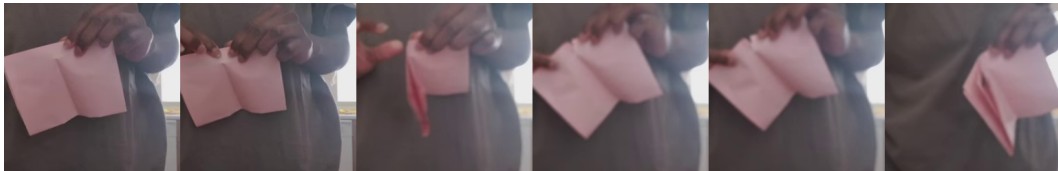

**Proficiency Test:** Someone tears the pink paper / curtain.
**Main Test:** Someone tears / assembles the pink paper.

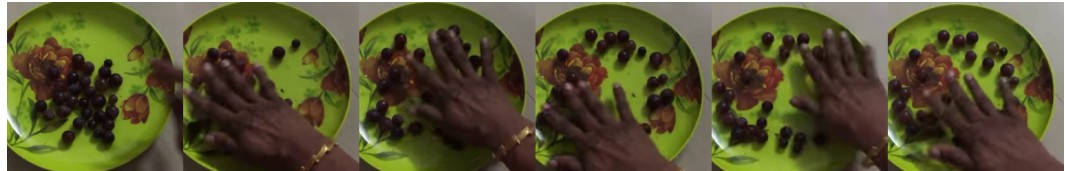

**Proficiency Test:** Someone spreads the grapes / word.
**Main Test:** Someone spreads / assembles the grapes.

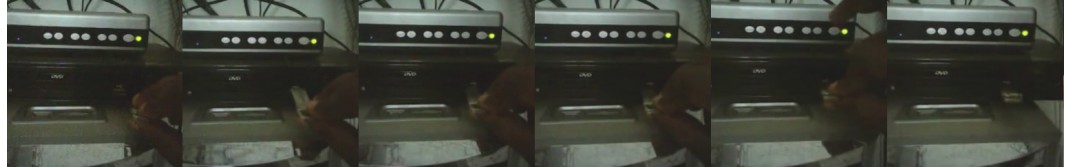

**Proficiency Test:** Someone attaches the pen drive / camera to the cd player.
**Main Test:** Someone attaches / detaches the pen drive to the cd player.

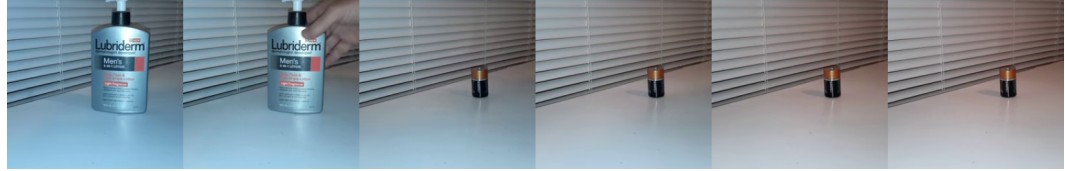

**Proficiency Test:** Someone reveals the battery / truth.
**Main Test:** Someone reveals / covers the battery.

Figure 16: Sample instances from the **Change of State (Action)** tests.

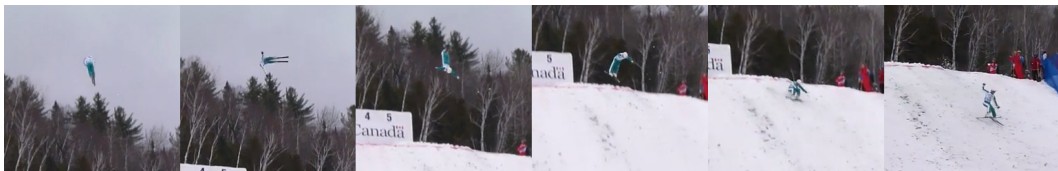

**Proficiency Test:** The skier / jet skies.
**Main Test:** Initially, the skier is in a higher / lower position.

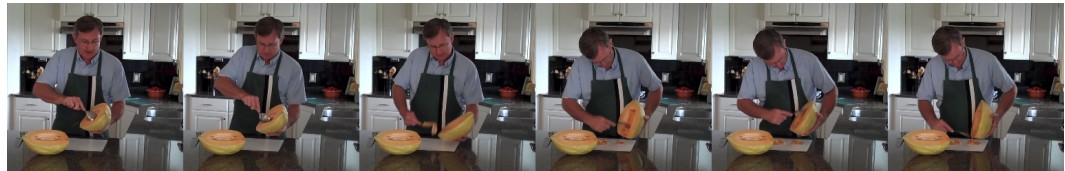

**Proficiency Test:** Someone digs the seeds / body out.
**Main Test:** Initially, the seeds are inside / outside.

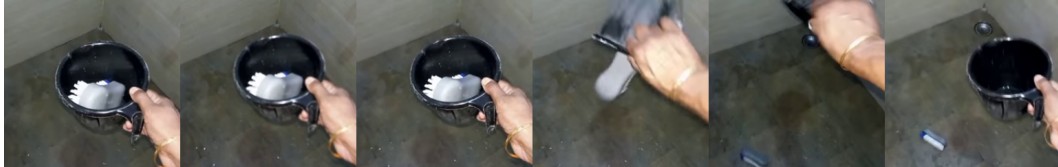

**Proficiency Test:** Someone empties the cup / trash.
**Main Test:** Initially, the cup is full / empty.

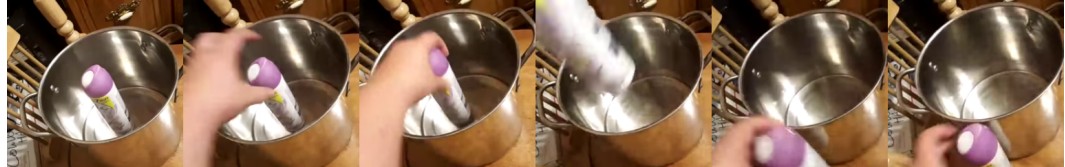

**Proficiency Test:** Someone pulls the spray / phone out.
**Main Test:** Initially, the spray is inside / outside.

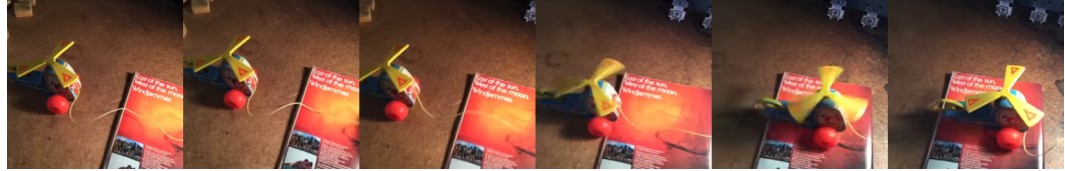

**Proficiency Test:** Someone pulls a baby baby toy / gun.
**Main Test:** Initially, a baby toy is further away / closer.

Figure 17: Sample instances from the **Change of State (Pre-State)** test.

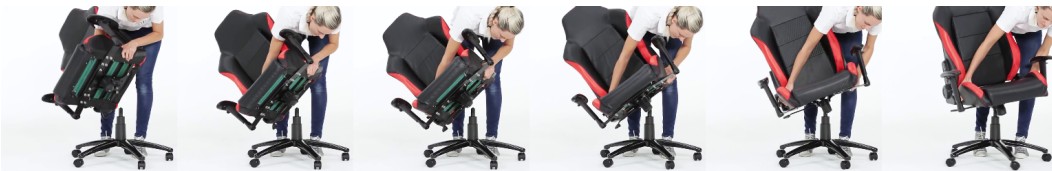

**Proficiency Test:** Someone connects the chair and the base / dots.
**Main Test:** At the end, the chair and the base is attached / detached.

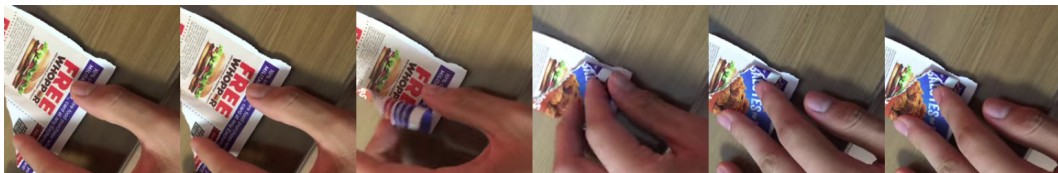

**Proficiency Test:** Someone folds the paper / laundry.
**Main Test:** At the end, the paper is folded / unfolded.

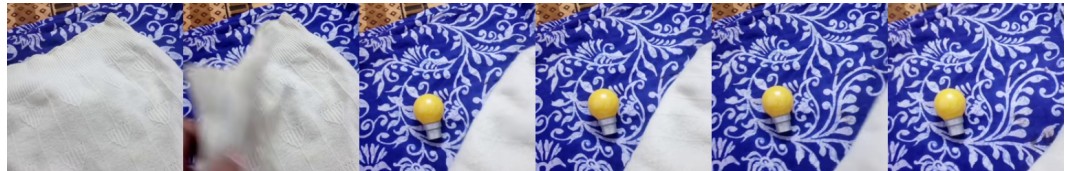

**Proficiency Test:** Someone uncovers the bulb / truth.
**Main Test:** At the end, the bulb is visible / hidden.

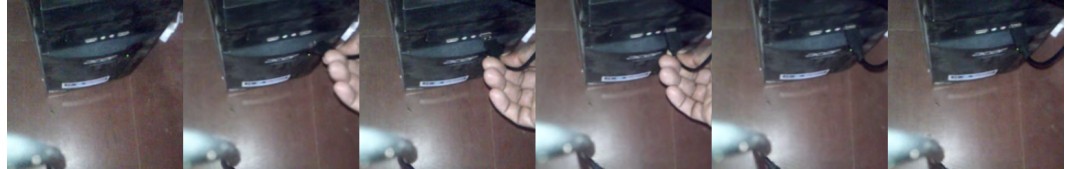

**Proficiency Test:** Someone plugs the usb / plug.
**Main Test:** At the end, the usb is inside / outside.

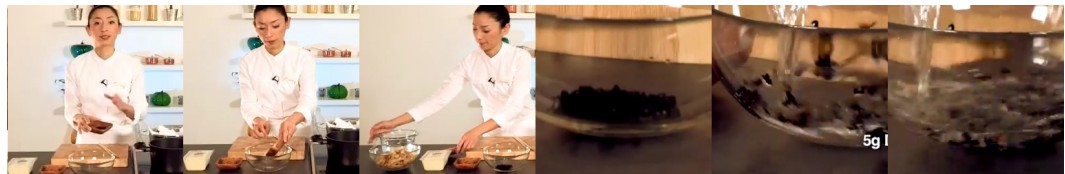

**Proficiency Test:** Someone soaks the wakame / car.
**Main Test:** At the end, the wakame is wet / dry.

Figure 18: Sample instances from the **Change of State (Post-State)** test.

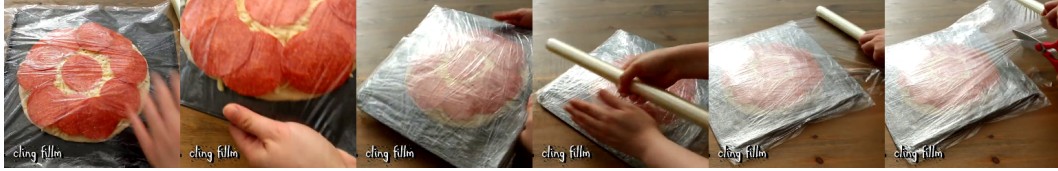

**Proficiency Test:** Someone wraps the pizza / presents.
**Main Test:** Initially, the pizza is unwrapped / wrapped. Then, someone wraps / unwraps the pizza. At the end, the pizza is wrapped / unwrapped.

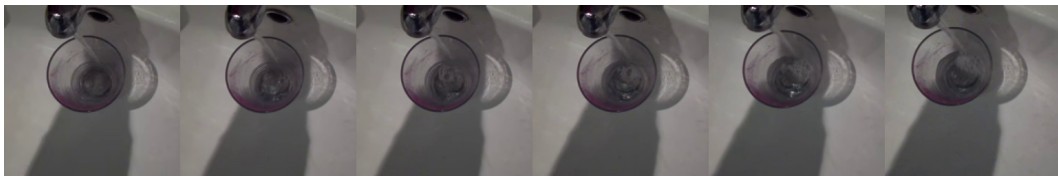

**Proficiency Test:** Someone fills the glass / gap.
**Main Test:** Initially, the glass is empty / full. Then, someone fills / empties the glass. At the end, the glass is full / empty.

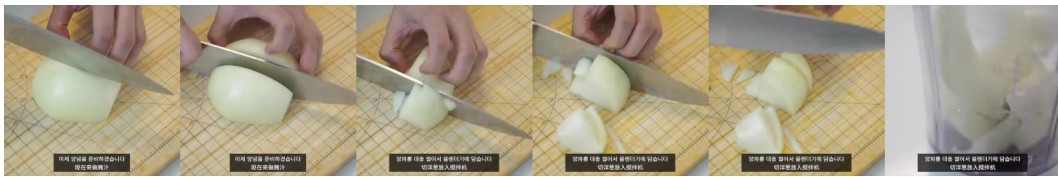

**Proficiency Test:** Someone chops an onion / apple.
**Main Test:** Initially, an onion is whole / in pieces. Then, someone chops / connects an onion. At the end, an onion is in pieces / whole.

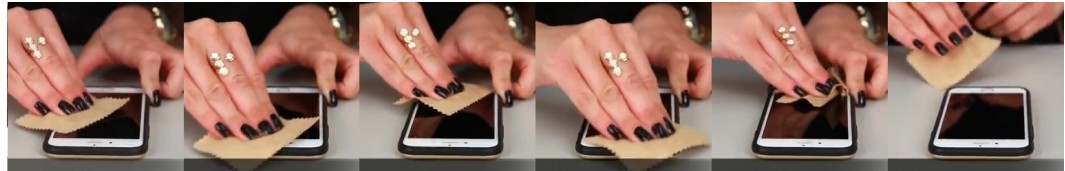

**Proficiency Test:** Someone wipes the screen / floor.
**Main Test:** Initially, the screen is dirty / clean. Then, someone wipes / stains the screen. At the end, the screen is clean / dirty.

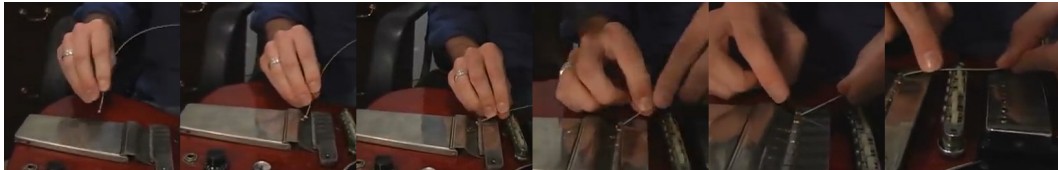

**Proficiency Test:** Someone fixes the new string / problem.
**Main Test:** Initially, the new string is separated / attached. Then, someone fixes / detaches the new string. At the end, the new string is attached / separated.

Figure 19: Sample instances from the **Change of State (Reverse)** test.

### C.5 RARE ACTIONS

In the **Rare Actions** test, we investigate the ability of VidLMs to identify novel compositions and recognise unusual events, such as a computer keyboard is being cut using a chainsaw by someone described. These events are described by a verb-noun pair, e.g. "*cutting a keyboard*". We choose foils from more likely events taking place in videos.

**Data sources.** We leverage the RareAct dataset (Miech et al., 2020), which consists of videos accompanied by action-object pairs describing events within the videos. These action-object pairs are extracted by analysing co-occurrence statistics from the widely used HowTo100M (Miech et al., 2019) dataset for VidLM pretraining. To enrich this dataset, we generate simple captions based on the action-object pairs. For instance, given the action-object pair *cut-keyboard*, we create the descriptive caption *cutting a keyboard*. We avoid subject phrases in captions since some videos do not comprise a human being as an actor.

**Foiling method.** This test offers two subtests: the **Action Replacement** and the **Object Replacement** subtest. In the Action Replacement subtest, we substitute the original action with a more plausible alternative that can be applied to the given object, e.g. *type on* for the previous *keyboard* example. To generate foils in this subtest, we employ T5 (Raffel et al., 2020), as it can produce foil candidates with *compound verbs*, e.g., *talk on*, *place at*, etc. We discard foil candidates with some general actions (e.g. use, have etc.) or actions that imply some form of *touching* (e.g. hold, reach etc.). We then perform an NLI and GRUEN score filtering. In this stage, we perform a manual intervention and abandon the low quality candidates. As for the Object replacement subtest, we focus on replacing the object in the action-object pair. For instance, revisiting the previous example, we replace the object *keyboard* with *bread*. Here, we prefer to use a set of token-based MLMs (Devlin et al., 2019; Lan et al., 2020; Liu et al., 2019). To further enhance the quality of the foils, we opt for an ensembling approach in the object replacement test. Particularly, we use three MLMs which are BERT, RoBERTa and ALBERT (Devlin et al., 2019; Liu et al., 2019; Lan et al., 2020). We also run an object detector called End-to-End Object Detection model (DETR) (Carion et al., 2020) and discard the foil candidates that contain the detected objects. We use `facebook/detr-resnet-101` DETR in Huggingface's transformers package (Wolf et al., 2019). We uniformly sample $K = 8$ frames and run the object detector on these sampled frames. We classify an object detected if its confidence threshold exceeds $0.80$.

**Proficiency Test.** The proficiency test of the Rare Actions focuses on the existence of objects in the given input videos, similar to the VALSE existence instrument. This time we do not use negated foils: We replace the correct object with another. To create captions, we create a statement about the existence of the ground truth object, e.g. "*there is at least one keyboard.*". To create foils, we randomly sample from the objects appear in ground-truth captions, e.g. "*there are some flowers*". Similar to the main test, we implement the same object detection filtering process.

**In-Depth Results.** Table 10 presents the model performance achieved on the Rare Actions test.

UNIMODAL RESULTS. Unimodal baselines GPT-2 and OPT (Radford et al., 2019; Zhang et al., 2022) perform very poorly on the main tests as expected since the captions describe less likely events. We observe that the models consistently perform better in object replacement tests in comparison to action replacement. This is inline with previous work where the models are more biased towards nouns and they fail to process verbs sufficiently (Momeni et al., 2023; Park et al., 2022; Lei et al., 2022).

IMAGE-LANGUAGE MODEL RESULTS. CLIP consistently outperforms random in all individual and combined tests, showcasing its effectiveness in understanding and executing the given tasks. In contrast, BLIP2, while excelling in proficiency tests, demonstrates lower performance in the main test and combined test compared to CLIP, implying potential limitations or challenges in action replacement subtest for BLIP2. Additionally, CLIP outperforms all VidLMs except VindLU and InternVideo, which demonstrates its ability to handle novel compositions is superior to these models.

Table 10: Rare Actions subtest results using pairwise accuracy ($acc_r$) metric. P, T and P+T stand for the scores achieved on the proficiency tests, the main tests only and the combined tests, respectively.

| Model | Action Replacement | | | Object Replacement | | | All | | |
|---|---|---|---|---|---|---|---|---|---|
| | P | T | P+T | P | T | P+T | P | T | P+T |
| Random | 50.00 | 50.00 | 25.00 | 50.00 | 50.00 | 25.00 | 50.00 | 50.00 | 25.00 |
| GPT-2 | 61.25 | 17.98 | 9.85 | 55.20 | 34.39 | 26.16 | 58.35 | 25.85 | 17.67 |
| OPT | 60.99 | 20.51 | 10.79 | 56.79 | 27.60 | 19.36 | 58.97 | 23.91 | 14.90 |
| CLIP | 92.81 | 93.74 | 86.95 | 92.63 | 94.08 | 88.73 | 92.72 | 93.90 | 87.80 |
| BLIP2 | 93.21 | 64.98 | 60.19 | 94.51 | 84.83 | 81.65 | 93.83 | 74.50 | 70.48 |
| ClipBERT | 38.75 | 27.96 | 7.99 | 40.75 | 52.60 | 20.81 | 39.71 | 39.78 | 14.14 |
| UniVL | 77.23 | 78.83 | 60.59 | 77.75 | 77.17 | 59.25 | 77.48 | 78.04 | 59.89 |
| VideoCLIP | 84.15 | 75.10 | 62.98 | 83.82 | 80.64 | 72.40 | 83.99 | 77.75 | 67.50 |
| FiT | 89.21 | 86.55 | 76.30 | 90.17 | 92.49 | 85.55 | 89.67 | 89.40 | 80.73 |
| CLIP4Clip | 82.95 | 93.07 | 78.16 | 82.94 | 95.23 | 79.19 | 82.95 | 94.10 | 78.65 |
| VIOLET | 86.68 | 85.49 | 73.50 | 87.57 | 87.86 | 75.87 | 87.10 | 86.63 | 74.64 |
| X-CLIP | 82.69 | 86.28 | 70.57 | 85.12 | 84.97 | 74.13 | 83.85 | 85.65 | 72.28 |
| MCQ | 91.48 | 86.82 | 79.09 | 91.18 | 90.75 | 85.69 | 91.34 | 88.70 | 82.26 |
| Singularity | 92.14 | 85.22 | 78.16 | 93.21 | 92.77 | 88.44 | 92.65 | 88.84 | 83.09 |
| UniPerceiver | 57.12 | 55.12 | 31.02 | 59.39 | 62.71 | 38.72 | 58.21 | 58.76 | 34.71 |
| Merlot Reserve | 83.75 | **96.27** | 80.42 | 83.81 | 84.39 | 74.56 | 83.78 | 80.57 | 77.61 |
| VindLU | 94.01 | 92.14 | 86.42 | 94.36 | 94.08 | 89.60 | 94.18 | 93.07 | 87.94 |
| InternVideo | **95.47** | 95.87 | **92.65** | **95.81** | **97.54** | **92.65** | **95.63** | **96.67** | **92.65** |
| mPLUG-2 | 52.50 | 43.10 | 21.70 | 49.00 | 51.30 | 26.50 | 50.80 | 47.00 | 24.00 |
| Otter | 56.06 | 58.85 | 30.89 | 56.21 | 58.67 | 40.32 | 56.14 | 58.76 | 35.61 |
| Video-LLaMA | 78.70 | 58.19 | 47.00 | 78.76 | 84.97 | 71.10 | 78.73 | 71.01 | 58.56 |

VIDEO-LANGUAGE MODEL RESULTS. All VidLMs, consistently outperform random in all individual and combined tests, showcasing their effectiveness in understanding and performing the given tasks. The high accuracy scores across proficiency, main, and combined tests suggest the capability of these models in achieving accurate results in Rare Actions subtests. However, UniPerceiver struggles with accurately comprehending and replacing actions or objects in the given scenarios compared to other VidLMs, potentially indicating a need for improvements in its ability to understand nuanced visual and contextual information.

**Test Examples.** In Figure 20 and Figure 21, we show some sample validated examples from the **Rare Actions – Action Replacement and Object Replacement** subtests.

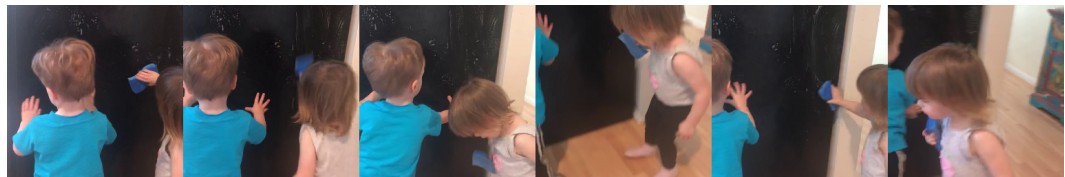

**Proficiency Test:** there is at least one fridge / chocolate
**Main Test:** washing some fridges / eating from a fridge

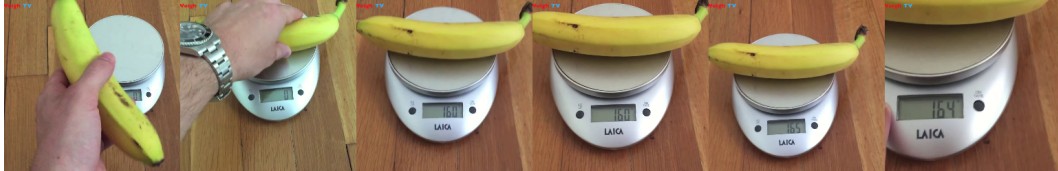

**Proficiency Test:** there is at least one banana / blender
**Main Test:** weighing / eating a banana

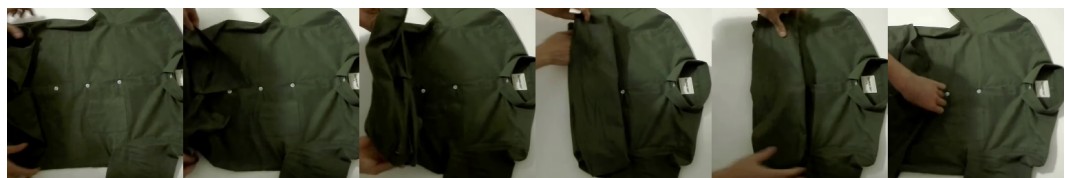

**Proficiency Test:** there is at least one shirt / are some peppers
**Main Test:** rolling / putting a shirt

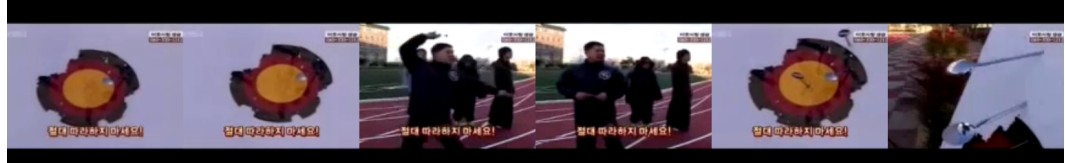

**Proficiency Test:** there are some spoons / is at least one car
**Main Test:** throwing / serving with some spoons

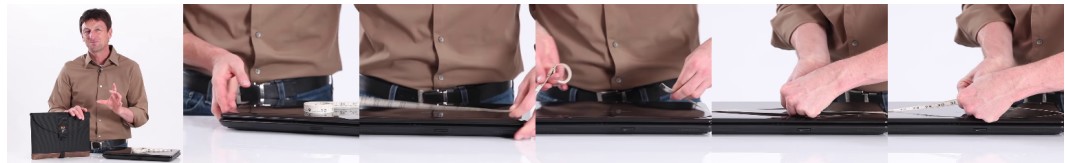

**Proficiency Test:** there is at least one laptop / car
**Main Test:** measuring / accessing a laptop

Figure 20: Sample instances from the **Rare Actions (Action Replacement)** test.

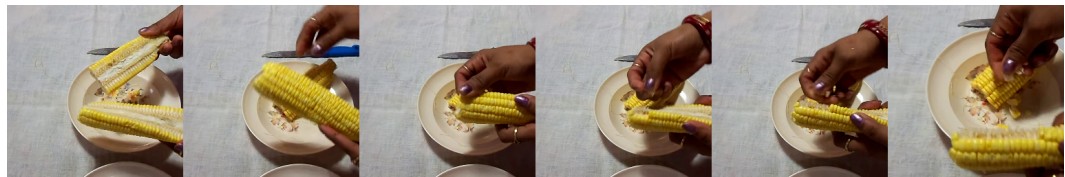

**Proficiency Test:** there are some corns / is at least one blender
**Main Test:** peeling some corns / a lemon

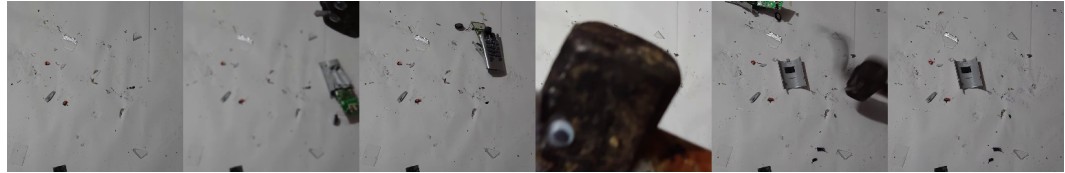

**Proficiency Test:** there is at least one phone / egg
**Main Test:** hammering a phone / some nails

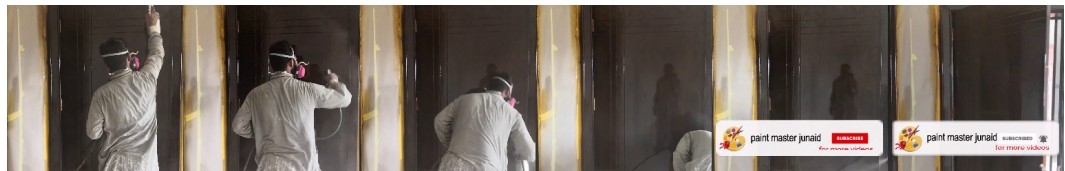

**Proficiency Test:** there is at least one door / towel
**Main Test:** spraying a door / an area

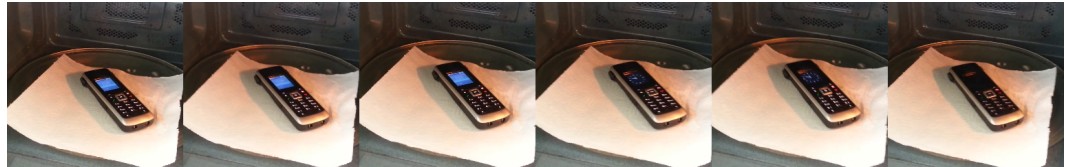

**Proficiency Test:** there is at least one phone / bicycle
**Main Test:** microwaving a phone / some food

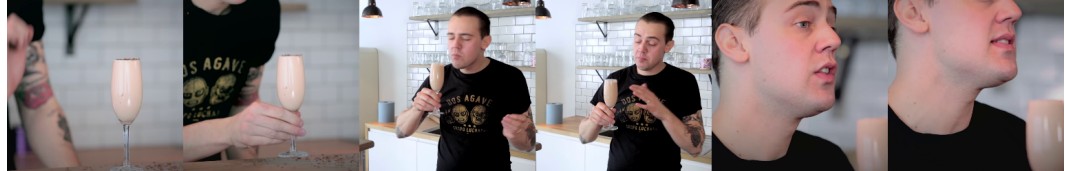

**Proficiency Test:** there is at least one chocolate / are some corns
**Main Test:** drinking a chocolate / some tea

Figure 21: Sample instances from the **Rare Actions (Object Replacement)** test.

## C.6 Spatial Relations

In the Spatial Relations test, we investigate the capabilities of VidLMs to understand spatial as well as spatio-temporal relations in a video (e.g. moving something *towards* or *from* something).

**Data sources**   We create the foils starting from the Something-Something V2 dataset Goyal et al. (2017a), which contains a collection of $220,847$ labelled clips of 174 pre-defined basic actions performed by humans with common objects, such as *putting something into something* or *turning something upside down*. Since this dataset is often used to pretrain VidLMs models, we leverage only the video-caption pairs present in the validation set.

**Foiling method.**   In order to create a foil, we generate several candidates for each caption by replacing a preposition in the caption with others drawn from the set of prepositions in the validation set itself. This ensures that foils and captions express relations with prepositions from the same distribution. Each candidate is scored for perplexity using *T5-base*[15]. We test also with *T5-large*, but we obtain similar results, therefore we use the base version for better efficiency.

In order to facilitate the model in the scoring stage, we prepend a subject to the candidate before feeding it into the model. Therefore a sentence like, *rolling a can onto a flat surface* becomes *someone rolling a can onto a flat surface*. We observe that with this format, the model produces better perplexity scores, probably because the data T5 was trained on tends to contain full sentences with explicit subjects, rather than verb phrases only. However, this is carried out only for scoring purposes. Once a candidate is selected, the version included in the main test does not have any prepended subject.

We select the top 10 best-scoring candidates for each sample and we run them through a state-of-the-art NLI model based on RoBERTa[16], using the caption as premise and the candidate foil as a hypothesis. Similarly to Parcalabescu et al. (2022), we keep only the candidates predicted as *neutral* or *contradiction*. The candidate foils are checked for grammaticality, by computing the GRUEN score and filtering out candidates with scores lower than $0.6$. Finally, we adjust the distribution of prepositions in the foils to match the caption distribution. This is achieved by removing foil candidates containing prepositions with frequencies that exceed the frequency of the caption distribution. This mitigates distribution biases arising in the foil generation process, which could be exploited by the tested models.

**Proficiency Test.**   We focus on the **object identification** task for the proficiency test of **Spatial Relations**, similar to Change of State proficiency tests. We use RoBERTa (Liu et al., 2019) to generate the foil by either masking the subject or the object of the caption, again, depending on the transitivity of the verb as we did in Change of State.

**In-Depth Results.**   We report in Table 11 the results for **Spatial Relations**.

UNIMODAL RESULTS.   Unimodal models, namely GPT-2 and OPT show higher performance on the main test (T) than on the proficiency test (P) where they are close to random chance. The quite decent result on the main test (T) suggests that the models can still exploit some spurious correlations in the text. This may be due to the design choice to include only in-distribution relations in the foil construction which although, filtered for grammaticality may still be detected as less likely than the actual caption. This can certainly be seen as a potential limitation of the Spatial Relations test. However, this is less visible in the combined test (P+T) where both perform better than the random baseline.

IMAGE-LANGUAGE MODEL RESULTS.   Both CLIP and BLIP2 perform well on the proficiency test (P). This is expected as this test heavily relies on object identification and it is known that ILMs are usually trained with object-centric texts. On the other hand, we observe a drop in performance on the main test (T) for both models and, to a greater extent on CLIP. This is also expected since the task is designed to exploit temporal information, which cannot be exploited by these models.

---

[15] https://huggingface.co/t5-base
[16] https://huggingface.co/ynie/roberta-large-snli_mnli_fever_anli_R1_R2_R3-nli

Table 11: Spatial Relations results using pairwise accuracy ($acc_r$) metric. P, T and P+T stand for the scores achieved on the proficiency tests, the main tests only and the combined tests, respectively.

| Model | Spatial Relations | | |
|---|---|---|---|
| | P | T | P+T |
| Random | 50.0 | 50.0 | 25.0 |
| GPT-2 | 49.1 | 72.8 | 43.0 |
| OPT | 59.0 | 84.7 | 55.7 |
| CLIP | 78.6 | 58.3 | 44.8 |
| BLIP2 | **91.1** | 86.0 | 79.4 |
| ClipBERT | 44.0 | 65.1 | 30.0 |
| UniVL | 62.5 | 51.7 | 33.2 |
| VideoCLIP | 67.9 | 54.7 | 39.7 |
| FiT | 70.5 | 51.9 | 38.7 |
| CLIP4Clip | 79.8 | 56.7 | 44.2 |
| VIOLET | 73.3 | 50.4 | 38.7 |
| X-CLIP | 74.8 | 56.2 | 43.5 |
| MCQ | 79.4 | 48.9 | 39.4 |
| Singularity | 80.7 | 46.8 | 38.9 |
| UniPerceiver | 45.5 | 48.0 | 20.1 |
| Merlot Reserve | 63.1 | 41.9 | 29.2 |
| VindLU | 83.2 | 45.6 | 39.4 |
| InternVideo | 76.6 | 59.8 | 45.3 |
| mPLUG-2 | 46.6 | 48.1 | 26.5 |
| Otter | 62.9 | 71.3 | 47.6 |
| Video-LLaMA | 88.6 | **88.8** | **79.6** |

Despite that, BLIP2 keeps performing decently in both the main test (T) and the combined one (P+T), setting the best performance on the spatial relations among the evaluated models. This raises questions regarding the capability of VidLMs to ground visio-temporal and textual information.

VIDEO-LANGUAGE MODEL RESULTS. Apart from UniPerceiver and ClipBERT (which perform below chance), all the VidLMs perform decently on the proficiency test (P). However, their performance consistently drops for both the main (T) and the combined test (P+T). Video-LLaMA is the best-performing VidLM. That being said, the performance of the remaining models is far from both unimodal and ILMs. This suggests a consistent lack of grounding in VidLMs concerning spatial relations and the need for VidLMs to design better strategies to properly leverage the temporal information in video-language tasks.

**Test Examples.** In Figure 22, we show some sample validated examples from the **Spatial Relations** test.

## D FURTHER ANALYSIS

In Figure 23, we present two subplots depicting pertinent insights regarding model performance in relation to video length. The left subplot illustrates the dynamics of model accuracy across various video duration segments. The data comprises six distinct time groups, each corresponding to the performance metrics of ten different VidLMs. Conversely, the right subplot visually represents the

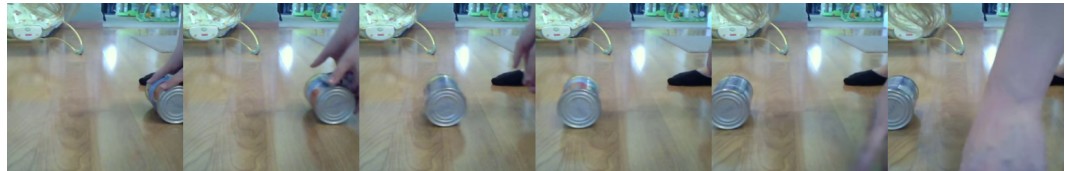

**Proficiency Test:** rolling a can on a flat surface / screen
**Main Test:** rolling a can on / onto a flat surface

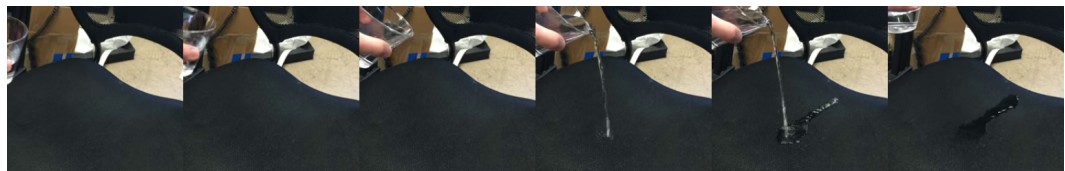

**Proficiency Test:** pouring water onto a chair / table
**Main Test:** pouring water onto / out of a chair

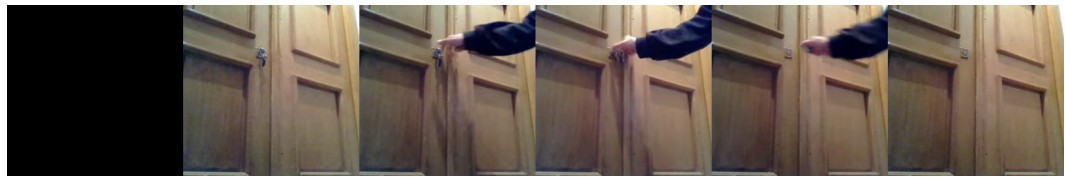

**Proficiency Test:** pulling keys out of a lock / drawer
**Main Test:** pulling keys out of / on a lock

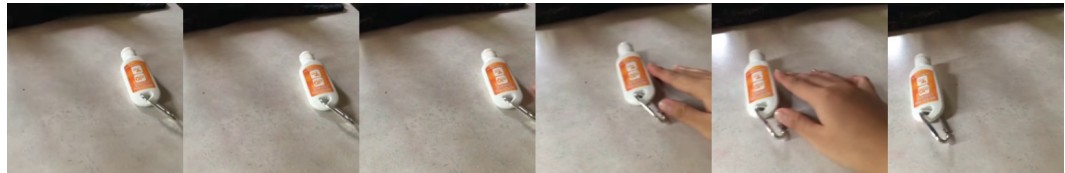

**Proficiency Test:** pushing small sunscreen lotion from right to left / bottom
**Main Test:** pushing small sunscreen lotion from / on right to left

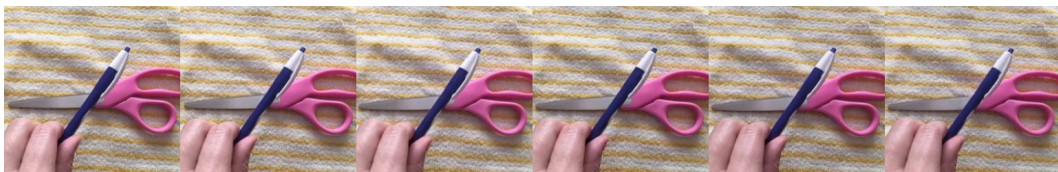

**Proficiency Test:** holding a pen over scissors / paper
**Main Test:** holding a pen over / with scissors

Figure 22: Sample instances from the **Spatial Relations** test. The descriptions are quite similar except for a small lexical variation (in blue) for the caption and (in orange) for the foil.

distribution of samples categorised within the aforementioned time groups. This graphical depiction provides an intuitive understanding of how the dataset is distributed across different video length categories. Our preliminary analysis suggests that the majority of the implemented VidLMs perform worse as the video length increases. The reason behind this issue is how these models process input videos, where many of them uniformly sample a limited number of frames and discard time information at the same time.

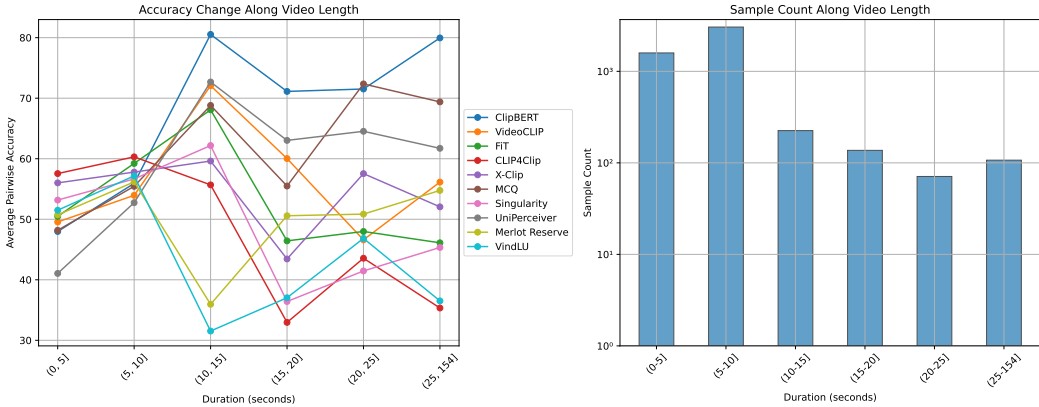

Figure 23: Analysis of performance variation across video length (left) and distribution of samples based on video length in log-scale (right).

