# OpenReview forum: "ViLMA: A Zero-Shot Benchmark for Linguistic and Temporal Grounding in Video-Language Models"
_ICLR.cc/2024/Conference — ICLR 2024 poster_

### Official Review · Reviewer_DXPK · 2023-10-20

**Soundness:** 3 good
**Presentation:** 3 good
**Contribution:** 2 fair
**Rating:** 6
**Confidence:** 2

**Summary:**

This paper proposed a zero-shot benchmark for linguistic and temporal grounding in video-language models. The evaluation focuses on five aspects: action counting; the recognition of specific actions or action participants; the recognition of action or event subphases; the recognition of rare actions; and distinguishing spatial relations. Experiments show that there is no essential difference between video-language models and image-language models in terms of temporal reasoning abilities.

**Strengths:**

1. The motivation of this paper is very important, that is, to establish a fair and reasonable benchmark for linguistic and temporal grounding in video-language models.
2. The experiments in this paper reveal that there is no essential difference between video-language models and image-language models in terms of temporal reasoning abilities. It provides the direction for the future development of the video-language models.
3. The paper is well written and easy to follow.

**Weaknesses:**

1. The form of the benchmark is still relatively simple, just let models choose the correct answer from two candidate sentences. However, simple two-choice questions are not enough to fully measure the ability of the model.

2. In my opinion, the temporal understanding ability should include having the model locate where an event starts and ends in a video based on the description. More complex temporal understanding requires the model to analyze the events that occur in the video, and infer the actions that may occur in subsequent videos. Therefore, in my opinion, the proposed benchmark does not fully measure the temporal understanding ability of video-language models.

3. With the success of large language models, the latest video-language models, e.g., BLIP2 can output text with variable length and free content. The community may be more concerned about how to properly evaluate these open outputs.

**Questions:**

Do you consider doing more analysis on the video itself, such as exploring the sensitivity of the model's temporal understanding ability to the length of the video? What happens to the model if you insert noise frames into the video?

---

> ### Author Response · Authors · 2023-11-19
>
> We greatly appreciate your insightful feedback and the opportunity to clarify aspects of our work. Below, we address each of your concerns in detail.
>
> ### Simplicity of the Benchmark Formulation
> We understand your concern regarding the simplicity of our benchmark. However, we argue that this simplicity is intentional and beneficial. The two-choice question format was chosen to precisely evaluate specific capabilities of video-language models (VidLMs) without the confounding effects of more complex task structures. This straightforward approach allows us to directly assess whether a model possesses a particular capability, avoiding confusion that could arise from more intricate tasks. Moreover, our foiling setup offers several advantages: it is task-agnostic, requires no additional fine-tuning or specific heads for evaluation tasks, and allows direct assessment of multimodal pre-training benefits without the risk of shortcut learning. This simplicity also facilitates faster and more straightforward testing, making our benchmark more accessible and easily adoptable by the research community.
>
> ### Temporal Understanding in Video-Language Models
> Your suggestion regarding the assessment of models' temporal understanding, including locating event starts and ends, is insightful. In our benchmark, this is indirectly addressed in the Action Counting test, where models must recognize and sum up the repetitions of an event, requiring an understanding of the event's visual loop and its linguistic representation. Additionally, our Change of State test, particularly in its reverse subtest, challenges models to align specific visual sub-events with their textual descriptions. This requires the model to segment both the video and the text into sub-phases and align them accurately, implicitly assessing the model's ability to determine where sub-events start and end.
>
> ### Evaluating Multimodal Large Language Models
> We acknowledge the growing interest in multimodal large language models and think evaluating them in a generative setting is interesting. That said, our primary aim is to evaluate specific capabilities of VidLMs in a clear and straightforward manner, rather than tackling the broader, unsolved problem of natural language generation evaluation.
>
> ### Regarding Your Question on Video Analysis
> Your query about the model's sensitivity to video length and the impact of noise frames is intriguing. We are currently working on a length-based performance analysis, as preliminary observations suggest that performance deteriorates with longer videos. This is likely due to the sampling methods used by some VidLMs and the increased complexity of modeling longer temporal dependencies. Please, check the appendix of the updated manuscript to see our analysis (D. Further Analysis).

---

> ### Comment · Reviewer_DXPK · 2023-11-22
>
> I appreciate the responses given by the authors. I am satisfied with some of the answers provided. However, I also strongly advise the authors to rigorously evaluate existing multimodal large language models on the proposed benchmarks. In the revised version, I see that the proposed benchmark is a challenging benchmark for the existing multimodal large language models, such as Otter. If the authors could extend this work to a new benchmark for multimodal large language models, it would significantly amplify the impact of this study.
>
> Considering the efforts made by the author during the response period, I finally decided to raise the rating from 5 to 6.

---

### Official Review · Reviewer_9Dpj · 2023-10-22

**Soundness:** 2 fair
**Presentation:** 3 good
**Contribution:** 3 good
**Rating:** 6
**Confidence:** 4

**Summary:**

This paper proposes a zero-shot evaluation benchmark designed to require a strong temporal understanding of video-language models. The proposed benchmark is task-agonistic. They evaluate multiple video-language models and image-language models on the proposed benchmark. They find that video-language models do not have a significant advantage over image-language models, and the input understanding of the model is not robust.

**Strengths:**

1. The proposed benchmark is novel and fills a gap in video language model evaluation that tests the zero-shot temporal understanding and reasoning capabilities.
2. The required capabilities in the proposed benchmark are well-classified.

**Weaknesses:**

1. There are some of the most recent VidLMs with good performances missing in the evaluation. E.g. InternVideo [1], mPLUG2 [2], Uniformer v2 [3], etc. The necessity of the proposed dataset needs more evaluation to validate.

[1] InternVideo: General Video Foundation Models via Generative and Discriminative Learning
[2] mPLUG-2: A Modularized Multi-modal Foundation Model Across Text, Image and Video
[3] UniFormerV2: Spatiotemporal Learning by Arming Image ViTs with Video UniFormer

**Questions:**

1. When testing image-language models, which frame from the video is input to the model?
2. Next-QA [1] also requires the model to perform temporal understanding and reasoning. It seems the main difference between the proposed dataset and Next-QA is the format. Can the author explain the core challenges posed by the proposed dataset?

[1] NExT-QA: Next Phase of Question-Answering to Explaining Temporal Actions

---

> ### Author Response · Authors · 2023-11-19
>
> Thank you for your insightful comments. We appreciate your feedback and have addressed your concerns as follows:
>
> ### Missing Recent Models:
> We acknowledge the importance of including recent video-language models (VidLMs) in our evaluation. To this end, we have adapted the InternVideo and mPLUG2 models. These models will be featured in the final version of our paper. Regarding UniFormer-v2, we excluded it due to its inherent design as a vision-only model, lacking the necessary capabilities for language processing, which is crucial for our benchmark's objectives. In the following, we present the results of InternVideo and mPLUG2 models (P+T). We will update our manuscript accordingly.
>
> |Method|Action Counting|Situ. Awareness|Change of State|Rare Actions|Spatial Relations|
> |-|-|-|-|-|-|
> |InternVideo|48.7|29.5|55.1|92.7|45.3|
> |mPLUG-2|27.7|21.5|20.8|24.0|26.5|
>
> ### Comparison to NExT-QA:
> Our benchmark differs from NExT-QA in several fundamental aspects.
> 1. Unlike NExT-QA, which primarily focuses on event recognition and localization, our benchmark is designed to challenge pretrained VidLMs with minimally edited counterfactual captions. This novel approach assesses various linguistic and spatio-temporal capabilities, such as understanding relational dynamics, counting repetitive actions, and observing state changes.
> 2. Our benchmark employs a counterfactual caption generation technique that provides a more nuanced and challenging test of model capabilities. This stands in contrast to NExT-QA’s multiple-choice format, where incorrect choices are often drawn from unrelated questions, leading to irrelevant options. For instance, consider this example from the NextQA paper:
> “Q: Why did the girl in blue stop and turn around at the start? Options: 0. Pour ingredient in, 1. Scared, 2. Waiting for the lady, 3. Take out vegetable, 4. Dance” (Correct Answer: 2).
> This format can unintentionally simplify the task for models by including distinctly irrelevant options, thus diminishing the quality of the assessment.
> 3. Our benchmark is specifically tailored as a zero-shot, test-split-only foiling benchmark, eliminating the need for additional finetuning or task-specific prediction heads. This design allows for a direct assessment of the visio-linguistic competencies acquired by models during their pre-training phase, thus providing a novel and robust measure of temporal understanding and reasoning in video-language models.
>
> ### Image-Language Model Input:
> In our methodology, we sample multiple frames across a video clip. For each frame, using the image-language model, we compute individual scores, which we then average to derive a final score for the whole video clip. Further details of this process are given in Section A.4, Implementation Details, of our paper.

---

> > ### Comment · Reviewer_9Dpj · 2023-11-22
> >
> > Thanks to the reviewer for the response. Given the results and the explanations, the reviewer thinks it is a meaningful dataset for video-language models.

---

### Official Review · Reviewer_cNVC · 2023-10-23

**Soundness:** 3 good
**Presentation:** 3 good
**Contribution:** 3 good
**Rating:** 6
**Confidence:** 3

**Summary:**

This paper studies the problem of evaluating the temporal understanding ability of Video Language Models (VidLMs). It proposes a benchmark called VILMA by constructing “foil” video captions from existing datasets. Specifically, the foil captions are created by replacing certain phrases in the original captions and then the VidLMs are asked to distinguish between the original and foil captions. The foil captions can be divided into five categories, covering a wide range of temporal understanding abilities. This paper also introduces the **proficiency tests**, which assess the primary abilities required to effectively understand the temporal dynamics (**main tests**). The proficiency tests are designed to examine whether the performance in main tests is robust.

Based on VILMA, a number of VidLMs, image-language models (ILMs) and text-only models are tested. The results show that: (1) existing VidLMs exhibit very poor temporal understanding ability, which is not better than ILMs (even not better than random baseline in particular categories). (2) The performance of VidLMs and ILMs declines significantly when considering the proficiency test, which suggests that they may predict correct answers by chance or by exploiting some spurious features.

**Strengths:**

* The proposed benchmark is novel and valuable, which can provide a more comprehensive evaluation of temporal understanding ability than existing benchmarks.
* The evaluation results reveal the poor temporal understanding ability of existing VidLMs struggle, which can guide the development of more advanced VidLMs.
* The paper provides comprehensive details of the proposed benchmark, including the construction process, data distribution and examples.

**Weaknesses:**

* The difference between VILMA and existing foiling benchmarks is not adequately described. The reviewer would like to know more details about why VILMA is more comprehensive.
* The gap between P+T suggests that there exists inherent dataset bias in VILMA, which can be exploited to achieve good performance. This phenomenon seems contradictory to the claim on page 24 that “the biases are not significantly present”.
* The evaluation could be more comprehensive by including recent Video Large Language Models (Video LLMs), e.g., VideoChat [1], Otter [2], Video-LLaMA [3].

[1] VideoChat: Chat-Centric Video Understanding.
[2] Otter: A Multi-Modal Model with In-Context Instruction Tuning.
[3] Video-LLaMA: An Instruction-tuned Audio-Visual Language Model for Video Understanding.

**Questions:**

Please refer to the **Weaknesses**

---

> ### Author Response · Authors · 2023-11-21
>
> Thank you for your detailed and insightful comments on our work. We highly value your feedback and are pleased to provide detailed clarifications on the points you raised.
>
> ### Distinctiveness of ViLMA from Existing Foiling Benchmarks:
> ViLMA stands out due to its comprehensive approach compared to existing video-language foiling benchmarks. Unlike Contrast Sets (Park et al., 2022), which focuses primarily on recognizing entities and actions, and Test of Time (Bagad et al., 2023), which concentrates on assessing the temporal order of events using synthetic video-text pairs, ViLMA encompasses a broader range of linguistic and temporal phenomena. This includes spatial relations, counting repeated actions, and observing state changes. Additionally, we test a diverse array of models to ensure robustness. Moreover, VALSE (Parcalabescu et al., 2022), while being the most similar to our benchmark, is designed exclusively for static Image-Language Models (ILMs). ViLMA, on the other hand, extends this scope by focusing on Video-Language Models (VidLMs) with an emphasis on temporal grounding. This distinction is critical for advancing the field, as it addresses a wider and more complex set of challenges in video-language understanding.
>
> ### Explaining the Performance Gap Between T and P+T:
> The performance gap between T and P+T is attributable to a plausibility bias, as also noted by Reviewer f3MM. Our methodology to mitigate this bias is outlined in our response to f3MM. In particular, we have taken careful measures to ensure that our human validation process does not reintroduce lexical distributional shifts between captions and foils. This is achieved by aligning the lexical distributions of captions and foils, as indicated in Appendix B.1, Bias Check (pages 22-24 of the revised version).
>
> ### Incorporation of Recent Video-Language Models:
> Acknowledging the importance of including recent video-language models (VidLMs) in our evaluation, we have integrated all the methods you mentioned: We have successfully concluded all experiments related to the Otter model and are currently awaiting the completion of experiments for the other models. Their results, which demonstrate the efficacy and relevance of our approach, will be included in the final version of our manuscript. This inclusion will not only enhance the comprehensiveness of our study but also it will ensure that our findings remain relevant and applicable to current advancements in the field.

---

> > ### Comment · Reviewer_cNVC · 2023-11-22
> > **Response to Author Rebuttal**
> >
> > Thanks for taking the time to respond to my comments. I'm glad to see the additional results of Video LLMs. While the benchmark may contain potential lexical biases, the paper thoroughly examines this issue, and the construction process is deliberately crafted to alleviate these biases. My concerns are well addressed.

---

### Official Review · Reviewer_f3MM · 2023-10-30

**Soundness:** 3 good
**Presentation:** 3 good
**Contribution:** 4 excellent
**Rating:** 6
**Confidence:** 4

**Summary:**

This paper presents a new suite of benchmarks for video-language models (VidLMs), which requires the VidLMs to distinguish between factual and counterfactual descriptions of the videos. The benchmarks are further divided into a number of tests, including Action Counting, Situation Awareness, Change of State, Rare Actions, and Spatial Relations. Each test has an easy version (proficiency test) and a hard version (main test).

The paper further evaluates a large number of VidLMs, together with LMs and ILMs. Interestingly, even the blind LMs can achieve significantly higher-than-random accuracy on the Situational Awareness and Spatial Relations tests, which suggests strong linguistic prior. However, the VidLMs often perform worse than the blind LMs on the two tests. Further, the image-only ILMs often outperform the VidLMs.

**Strengths:**

Evaluations of large VidLMs pose significant challenges. As the paper noted, even if the models achieve good accuracy on the main test, it does not mean it can achieve high scores on the supposedly easier proficiency test. Hence, many test results create misleadingly high performance numbers that lead to the illusion of human-like performance, and contribute to concerns of existential risks.

This paper presents a solid step in rigorously evaluating VidLMs. The datasets are carefully designed and curated. The AMT protocols seem well thought over. The Action Counting test seems especially challenging.

**Weaknesses:**

The Situational Awareness tests and Spatial Relations tests seem to suffer from strong linguistic priors (despite mediocre VidLM performance). The Rare Actions tests are surprisingly easy, with VindLU achieving 88% for the difficult P+T condition.

The analysis is relatively cursory (even though it's called "in-depth results" in the appendices). The number of parameters and amount of training data of the VidLMs should be reported along with the results. Preferably the authors can say a few words about the model architectures. I understand the reader can track these down from the original papers, but doing so would require significant effort given the number of baselines. Having these meta-data would help the reader understand if the model strengths are derived from the model size, the training data, or the architecture design.

**Questions:**

Nil

---

> ### Author Response · Authors · 2023-11-20
>
> Thank you for your comprehensive review of our paper. Your insights have been instrumental in refining our approach, and we address your concerns as follows:
>
> ### Linguistic Priors in Situation Awareness and Spatial Relations Tests
> We recognize the importance of your concern regarding linguistic priors in the Situation Awareness (SA) and Spatial Relations (SR) tests. In our design of ViLMA, we have implemented several strategies to minimize such biases. This included deploying Natural Language Inference (NLI) filters and GRUEN score evaluations, as well as manual validation through Amazon Mechanical Turk (AMT). These measures were aimed at ensuring a balanced representation of lexical items in both factual and counterfactual captions to reduce the models' innate lexical biases. Despite these efforts, completely eliminating plausibility bias, particularly in SA and SR tests, is challenging. For example, creating counterfactual scenarios in SA tests often results in a lower thematic fit (e.g., consider the sentence from Table 1 for SA “A policeman holds a blond man against a wall”), a phenomenon observed in the perplexity patterns of unimodal models. This highlights the inherent challenges in these tests, where subtle biases arise from fine-grained differences in actor roles. Notably, VidLMs consistently scored lower than unimodal LMs and ILMs, indicating their struggle with integrating language with video and utilizing temporal information from videos. Importantly, when combining proficiency and main tests, we observed a significant reduction in the performance of unimodal models, suggesting that our approach effectively mitigates most plausibility biases.
>
> ### The Rare Actions Test Results
> In the Rare Actions (RA) test, we aimed to evaluate the robustness of VidLMs against unusual action-object compositions. Our findings showed a distinct difference in the performance of unimodal LMs and VidLMs. Unimodal LMs typically preferred counterfactual captions, a tendency reflecting a linguistic probability bias. In contrast, VidLMs showed a remarkable ability to correctly identify captions in scenarios deviating from standard linguistic patterns, suggesting their effective integration of visual information. This contrast in the performance is particularly illuminating, highlighting the VidLMs' capability to overcome language-side expectancy biases. The granularity of the RA test, focusing on action or object recognition in potentially out-of-distribution scenarios, serves as an important measure against modality collapse, where multimodal models might overly rely on linguistic cues.
>
> ### Additional Information on VidLMs
> In response to your suggestion, we enhanced our paper with detailed information on the VidLMs, including training data, and architectural designs, included in the appendices (please see Table 3 under Section A.3 Video-Language Models, we will also add parameter sizes soon). Clearly, this addition will provide readers with a comprehensive understanding of the factors influencing the varied performance of these models.
>
> We hope this clarifies the points you asked in your review.

---

### Comment · Area_Chair_HTAc · 2023-11-22

Hi Reviewers for paper #7649,

The authors have reponded to all your reviews. Pls read and reply to them.

Thanks,
AC

---

### Meta-Review · Area_Chair_HTAc · 2023-12-06

**Metareview:**

In terms of strengths, reviewers were unanimous that this work was well-motivated, novel, timely and addressed an important topic. The benchmark is signifcantly broader in task scope than previous comparable ones. The paper was well-written, with comprehensive details provided. A large number of models were benchmarked, with even more added as part of the rebuttal process.

In this AC's view, the inclusion, combination and comparison of both basic proficiency tests and the trickier balanced counterfactual foils is particularly interesting, and it's a nice methodological innovation with potentially broader and lasting impact on future benchmarking works more generally.

In terms of weaknesses, reviewers intially raised valid concerns/questions about the relatively simplicity of the task format, how open-ended outputs can be evaluated, some models not being included in the benchmarking, and differences from existing foiling benchmarks. Overall, the authors' rebuttal addressed these concerns adequately, with reviewers stating that their concerns were addressed and/or raising their score.

There are a few remaining issues, but these are relatively minor. One is that the analyses could have been more in-depth and comprehensive; much space in the main paper was used for task/methodological description. The other is that temporal benchmarking aspects, especially long-horizon understanding, could have been stronger. As it stands, the Action Counting task is primarily the one task that examines this, with some of the other tasks dealing with short or unclear temporal horizons.

Overall, this is a solid benchmarking paper, with reviewers unanimous that it falls on the positive side of the acceptance boundary. There are a few areas for improvement, but on balance, it's a very timely and well-executed piece of work that makes an important contribution to the ICLR community (and likely beyond).

**Justification For Why Not Higher Score:**

More thorough analyses could have been performed, with much space in the main paper used for describing the sub-task definitions/methodology. These could easily have been summarized in the main paper and detailed in supplementary materials. As a result, there is primarily one main insight, when the work could have been more insightful and impactful.

**Justification For Why Not Lower Score:**

Interesting and novel idea (use of counterfactuals across diverse tasks) for an important and timely topic (VidLMs), with impressive number of models benchmarked.

---

### Decision · Program_Chairs · 2024-01-16

Accept (poster)